# TopoWeaver-R1: Reinforcing Difficulty-Aware Topology Evolution in Multi-Agent Competition-Level Code Generation

## Abstract

Recent studies have shown that large language model (LLM)-driven multi-agent systems (MAS) are promising for addressing complex problems, with competition-level code generation as a representative domain. By emulating the collaboration among human programmers, these systems leverage predefined interaction topologies to achieve notable gains. However, such fixed structures introduce interaction redundancy and excessive token costs as task difficulty drops. While graph pruning and generation methods can produce sparser topologies, they remain static during inference, unable to adapt to execution feedback, and often converge to limited density ranges. To overcome these issues, we propose TopoWeaver-R1, a reinforcement learning–optimized MAS centered on an LLM orchestrator agent, which supports end-to-end evolutionary dynamic interaction topology generation. For each query, it infers agent roles and task difficulty, then constructs a task-adapted, density-aware layered directed acyclic graph (DAG) topology. The topology evolves via execution feedback and history, thereby improving the task-solving performance of the generated code. On three competition-level and two basic code datasets, TopoWeaver-R1 achieves state-of-the-art accuracy, with up to 14.6% higher accuracy, 13% lower density and 68% lower token cost than the strongest baseline. Our approach transitions multi-agent topologies from static designs to dynamic, feedback-driven evolutionary designs with fine-grained, difficulty-aware density control.

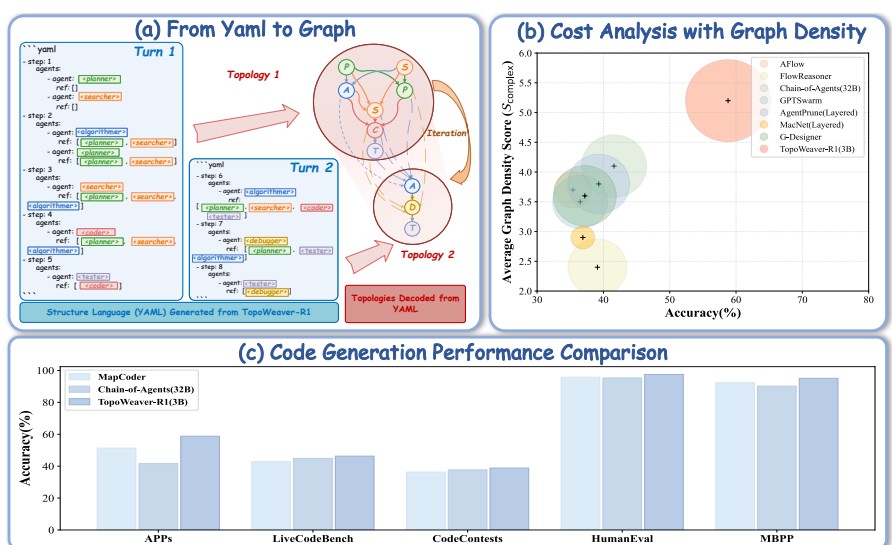

Figure 1: **(a)** YAML representation of the topology, its mapping to the actual graph, and the two-turn graph evolution. **(b)** APPS results showing performance, average graph density ($\mathcal{S}_{\text{complex}}\uparrow$ sparser), and completion tokens, with circle size indicating token savings (diameter↑ more). **(c)** Code generation performance comparison of representative baselines.

## 1 INTRODUCTION

Competition-level programming is widely regarded as one of the most demanding problem-solving tasks(Khan et al., 2023; Hendrycks et al., 2021). It requires deep understanding of problem statements, complex reasoning, algorithmic proficiency, and the ability to generate executable code that passes comprehensive test cases. While LLMs show strong general reasoning abilities, single models often lack the structural decomposition and iterative self-correction needed to solve such tasks effectively(Austin et al., 2021). LLM-based MAS have recently achieved remarkable progress in competition-level code generation(Islam et al., 2024; 2025). Their exceptional performance largely stems from carefully designed interaction topologies that facilitate efficient coordination. However, no fixed topology suits all problems: easy cases favor lean pipelines, while hard cases require denser, tightly coordinated interactions. Moreover, the topology usually remains fixed at inference, and execution feedback (unit test failures) does not induce structural changes. As a result, iterative execution may suffer from redundant interaction or degraded performance. This motivates a central question: ***How can we automatically generate task-specific interaction topologies that scale density with difficulty and evolve in response to execution feedback?***

A growing body of work has explored this direction. Graph pruning methods (Zhang et al., 2024a; Zhuge et al., 2024) reduce costs by iteratively removing edges or roles, but the resulting fixed topologies may not align with task-specific demands, leading to degraded performance. Graph-generation approaches (Zhang et al., 2024b) improve over pruning by conditioning on the input query, but they typically rely on monotonic sparsity constraints that drive convergence to a fixed density range, and the generated topology remains frozen during inference without adaptation to execution feedback. Workflow-centric RL methods (Gao et al., 2025; Li et al., 2025) train a single agent to manage linearized multi-stage workflows using end-to-end reinforcement learning. While effective under limited inter-agent overhead, they restrict interaction to sequential message passing and lack the expressiveness and adaptability of interaction graphs.

To realize these capabilities, we first introduce a novel layered DAG topology. Unlike prior designs(Qian et al., 2024), it enables intra-layer parallelism as well as cross-layer interactions. Distinctively, this topology is expressed in a structured language (YAML), which makes it human-readable and directly generable by LLM agents. Building on this foundation, we present **TopoWeaver-R1, a reinforcement learning(RL) optimized MAS centered on an LLM orchestrator agent that performs multi-turn, end-to-end dynamic generation of the above interaction topologies for competition-level code generation.** We first apply supervised fine-tuning(SFT) to equip the orchestrator with priors over interaction graphs. To better capture the characteristics of multi-agent interaction, we further propose a graph density evaluation function tailored to our proposed layered DAG structure. Finally, to optimize the orchestrator with RL, we design a multi-objective reward based on this metric that balances structural correctness, code accuracy, and density. A distinctive feature of our density reward is the introduction of difficulty-dependent bounds on topology density. This fine-grained control enables explicit cost–accuracy trade-offs under token budgets. In summary, our main contributions are as follows:

- We propose a **novel layered DAG topology for multi-agent interaction** that supports intra-layer parallelism and cross-layer interactions. The topology is represented in a human-readable format that can be directly generated by agents.

- We introduce **TopoWeaver-R1, an RL-optimized MAS centered on an LLM orchestrator agent**, which enables end-to-end difficulty-aware evolutionary dynamic interaction topology generation in competition-level code generation.

- We introduce a **graph density evaluation function for layered DAGs and use it to design a multi-objective reward function** balancing structural correctness, code accuracy, and difficulty-aware density under task-specific constraints.

- We demonstrate state-of-the-art performance on multiple competition-level and foundational code benchmarks, **achieving higher accuracy with lower average density and reduced cost compared to existing methods.**

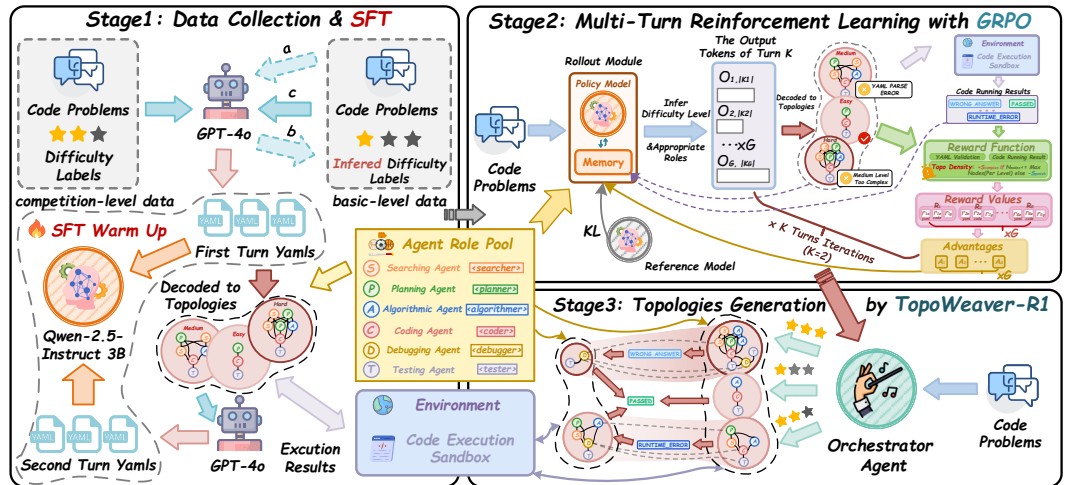

Figure 2: ***Overall framework of the proposed TopoWeaver-R1***. The approach proceeds in three stages: (1) SFT on diverse topologies to instill structural priors in the base LLM (Qwen-2.5-Instruct-3B); (2) RL with GRPO to learn task-adaptive, difficulty-aware topology policies from execution feedback, yielding the orchestrator agent; and (3) multi-turn dynamic topology generation for end-to-end code problem solving.

## 2 TOPOWEAVER-R1

TopoWeaver-R1 is an RL-optimized MAS centered on an orchestrator agent, designed for end-to-end, multi-turn generation of difficulty-aware evolutionary interaction topologies. The system is first SFT-trained on constructed topologies to instill structural priors, and then optimized with RL via GRPO to generate topologies that adapt to task difficulty and evolve through execution feedback. In this section, we present a detailed description of the overall framework and its components, as illustrated in Fig.2.

### 2.1 PROBLEM DEFINITION

#### 2.1.1 INTERACTION TOPOLOGY NOTATIONS

We first introduce a novel multi-agent interaction topology expressed in a human-readable structured language (YAML). As shown in Fig.1 (a), this topology is structurally defined as an improved layered DAG, where ***step*** denotes a layer and ***ref*** denotes an edge, supporting both intra-layer parallelism and cross-layer connections. Furthermore, it supports multi-turn evolutionary generation driven by execution feedback from multi-agent interactions. Formally, it is denoted as $\mathcal{G}^{(k)} = (\mathcal{V}^{(k)}, \mathcal{E}^{(k)})$, where $k$ is the turn index. Each node $v_i^{(k)} \in \mathcal{V}^{(k)}$ represents an agent instance that executes during turn $k$. The entire topology is generated and orchestrated by the orchestrator agent. See Appendix D for detailed notions of the interaction topology.

#### 2.1.2 TOPOWEAVER-R1 PARADIGM

Given a code problem $x$, the orchestrator agent policy $\pi_\theta$ generates, at turn $k \in \{1, \ldots, K\}$, a variable-length YAML token sequence

$$o_k = (o_{k,1}, \ldots, o_{k,|o_k|}), \tag{1}$$

that encodes the interaction topology. The sequence is deterministically decoded into a layered DAG

$$\mathcal{G}^{(k)} = \text{DecodeTopo}(o_k), \tag{2}$$

***In particular, TopoWeaver-R1 calibrates the topology density to the inferred difficulty of $x$. This induces variable $o_k$ lengths $|o_k|$ and reduces superfluous reasoning and token usage.*** The environment then executes agents according to $\mathcal{G}^{(k)}$ and returns feedback $z_k$ which can be further decomposed as $z_k = (z_k^{\text{roles}}, z_k^{\text{code}})$, where $z_k^{\text{roles}}$ collects the outputs of multiple agents generated, and $z_k^{\text{code}}$ denotes the sandboxed code-execution outcome. Let the turn history be $H_k = \{(\mathcal{G}^{(h)}, z_h)\}_{h<k}$. The joint process factorizes as

$$p_\theta(o_{1:K}, z_{1:K} \mid x) = \prod_{k=1}^{K} \underbrace{\pi_\theta(o_k \mid x, H_k)}_{\text{Topology generation}} \underbrace{P_{\text{env}}\left(z_k \mid x, \mathcal{G}^{(k)}, H_k\right)}_{\text{Execution feedback}}, \tag{3}$$

Equation 3 factorizes the multi-turn process into topology generation with environment execution: at turn $k$ the policy emits $o_k$ conditioned on $(x, H_k)$, the environment executes under $\mathcal{G}^{(k)}$ and returns $z_k$. Feedback $z_k$ is appended to $H_{k+1}$ and conditions the next generation, so the topology is updated online in response to execution feedback. See Appendix D.1 for algorithmic details.

### 2.1.3 GRAPH DENSITY EVALUATION FUNCTION

To better assess the complexity and performance of multi-agent interactions and explicitly account for cost consumption, we define the graph complexity evaluation function described by three metrics, including the number of nodes, the edge density and graph depth. The first two metrics can reflect the token costs, while the last indicator reflects the degree of parallelism of the system, or in other words, the response time. Let $n_i$ denote the number of agent invocations in step $i$, $s$ be the total steps for each round, then the total number of nodes is

$$|\mathcal{V}| = \sum_{i=1}^{s} n_i. \tag{4}$$

Edges are formed through agent references, with the total number of edges given by

$$|E| = \sum_{i=1}^{s} \sum_{j=1}^{n_i} |Agent_j[\text{ref}]|, \tag{5}$$

and the depth of the graph is related to the depth of invocation of the agent, denoted by $d$. Inspired by **Theorem 1**, we use the number of DAG layers (the total steps $s$) instead. For normalization, we map each metric into the unit interval $[0, 1]$. The normalized scores are defined as:

$$S_{\text{node}} = \exp\left(-\frac{|V|}{N_{\text{max}}(l)}\right), \quad S_{\text{edge}} = \exp\left(-\frac{|E|}{|V|(|V|-0.5)}\right), \quad S_{\text{depth}} = 1 - \frac{s}{|V|}. \tag{6}$$

where $l$ is task difficulty level, each level is associated with a maximum allowed number of nodes $N_{\text{max}}(l)$. $S_{\text{node}}$ reflects the node complexity based on the graph size. $S_{\text{edge}}$ captures the edge complexity relative to a complete graph, and $S_{\text{depth}}$ quantifies the spread of the graph by comparing its depth to the total number of nodes. The overall graph complexity evaluation function is defined as:

$$\mathcal{S}_{\text{complex}} = \alpha \cdot \exp\left(\lambda_1 \cdot S_{\text{node}} + \lambda_2 \cdot S_{\text{edge}} + \lambda_3 \cdot S_{\text{depth}}\right) \tag{7}$$

$\mathcal{S}_{\text{complex}}$ serves as a component of the reward function $r_\phi(\cdot)$, as defined in Eq.14, and contributes to the trajectory reward $\hat{A}_i$ in the Group Relative Policy Optimization (GRPO) advantage function, as detailed in Eq.8. The mathematical derivation that precisely defines $\mathcal{S}_{\text{complex}}$ as the topology density is provided in Appendix D.2.

### 2.2 SFT DATA GENERATION

To endow the base LLM with topology priors and facilitate its optimization during reinforcement learning, we built a supervised corpus. From three competition-level datasets and three difficulty tiers, we sampled 50 problems per tier per dataset (450 total). We designed a customized system prompt and queried GPT-4o to produce one YAML topology per problem. Each topology was validated by our checker for format correctness, de-duplication, and density within the difficulty band. For each topology, we constructed error-aware prompts from distinct failure types and generated a second-turn iterative topology. Combined with first-turn runs, this yielded 2,700 competition-level interaction graphs. We repeated the pipeline on two basic datasets to obtain 300 initial examples across difficulties; here the model inferred difficulty and generated the topology accordingly. In total we collected 4,500 examples. This produces a base model endowed with strong priors for topology generation.

Table 1: Rewards for Topology Validation and Code Execution Errors

| YAML Topology Correctness Rewards | | | Code Execution Error Rewards | | |
|---|---|---|---|---|---|
| **Error Type** | **Explanation** | **Reward** | **Error Type** | **Explanation** | **Reward** |
| [NO_YAML_FOUND] | No YAML block found. | -2.0 | [WRONG_ANSWER] | Code executes but outputs mismatch with expected | 1.0 |
| [YAML_PARSE_ERROR] | YAML parse failed. | -1.5 | [TIME_LIMIT_EXCEEDED] | Execution exceeded time limit. | 0.9 |
| [YAML_SCHEMA_INVALID] | YAML parsed, but fails the topology schema. | -1.0 | [MEMORY_LIMIT_EXCEEDED] | Execution exceeded memory limit. | 0.8 |
| [YAML_LOGIC_INVALID] | Violates topology logic rules. | -0.5 | [RUNTIME_ERROR] | Program crashed during execution. | 0.7 |
| ———— | ———— | - | [COMPILATION_ERROR] | Program failed to compile. | 0.6 |

## 2.3 REINFORCING DYNAMIC TOPOLOGIES FOR LLM-MA VIA TRAJECTORY-LEVEL POLICY OPTIMIZATION

**GRPO-Based Training for Dynamic Topology Generation**  After SFT, we further train the orchestrator policy to generate dynamic multi-agent interaction topologies using GRPO. See Appendix E.1 for the multi-turn trajectory and return definition. Specifically, the advantage of trajectory $i$ is defined as

$$\hat{A}_i = \frac{R_i(\tau) - \text{mean}\big(\{R_j(\tau)\}_{j=1}^G\big)}{\text{std}\big(\{R_j(\tau)\}_{j=1}^G\big)}, \tag{8}$$

Here, $R_i$ can be viewed as the instance-level realization of $R(\tau)$ (defined in Eq. 25) within the group of $G$ sampled trajectories.

The GRPO objective function can be formally expressed as follows:

$$J_{\text{GRPO}}(\theta) = \frac{1}{G} \sum_{i=1}^G \frac{1}{L_i} \sum_{k=0}^{K_i-1} \sum_{u=1}^{|o_{i,k}|} \min\left[ \frac{\pi_\theta(o_{i,k,u} \mid x, H_{i,k}, o_{i,k,<u})}{\pi_{\text{old}}(o_{i,k,u} \mid x, H_{i,k}, o_{i,k,<u})} \hat{A}_i, \right.$$
$$\left. \text{clip}\left( \frac{\pi_\theta(o_{i,k,u} \mid x, H_{i,k}, o_{i,k,<u})}{\pi_{\text{old}}(o_{i,k,u} \mid x, H_{i,k}, o_{i,k,<u})}, 1-\varepsilon, 1+\varepsilon \right) \hat{A}_i \right] - \beta\, \mathbb{D}_{\text{KL}}^{(\text{topo})}. \tag{9}$$

Here, $L_i = \sum_{k=0}^{K_i-1} |o_{i,k}|$ denotes the total number of topology tokens in trajectory $i$, $\varepsilon$ controls the clipping range, and $\mathbb{D}_{\text{KL}}^{(\text{topo})}$ is the token-level KL regularizer computed *only* over topology tokens (as in Eq. 26).

**Design of a Rule-Based Multi-Objective Reward Function**  The reward function directly influences the optimization process in RL. In this subsection, we elaborate on the definition of the immediate per-turn reward function $r_\phi(\cdot)$ introduced in Eq. 24.

To provide a single training signal that balances correctness, topology quality, and efficiency, we instantiate the immediate reward function in Eq. 24 as a weighted composite:

$$r_\phi(\mathcal{G}^{(k)}, z_k^{\text{code}}) = w_1\, r_e(\mathcal{G}^{(k)}, z_k^{\text{code}}) + w_2\, r_g(\mathcal{G}^{(k)}) \tag{10}$$

where the non-negative weights $w_i$ reflect the relative importance of each component. Here, $r_e$ (execution correctness) is derived from $z_k^{\text{code}}$ and $\mathcal{G}^{(k)}$, providing a reward for both the YAML validation and the code execution results; $r_g$ (graph density) evaluates the interaction topology $\mathcal{G}^{(k)}$, serving as the topology density reward function. This instantiation makes explicit that $r_\phi(\cdot)$ in Eq. 24 is realized as a weighted sum of multiple objectives, yielding a scalar reward signal for trajectory-level optimization.

**Execution Result Reward**  We first validate the format after the commander generates YAML. If no YAML is found or YAML does not match the rule, the system raises an error, and gives a punishment according to the type of error. The types of error are shown as:

$$\mathcal{E}_{\text{yaml\_errors}} = \{ [\text{NO\_YAML\_FOUND}], [\text{YAML\_PARSE\_ERROR}], [\text{YAML\_SCHEMA\_INVALID}],$$
$$[\text{YAML\_LOGIC\_INVALID}] \} \tag{11}$$

Then the testing agent gives the evaluation results of the generated code. Unless the result of test case matches the expected answer, the system raises a fail information based on the code run results. The error types for the code execution are defined and summarized as follows:

$$\mathcal{E}_{\text{code\_errors}} = \{\boxed{\texttt{[WRONG\_ANSWER]}}, \boxed{\texttt{[TIME\_LIMIT\_EXCEEDED]}}, \boxed{\texttt{[MEMORY\_LIMIT\_EXCEEDED]}}, \\ \boxed{\texttt{[RUNTIME\_ERROR]}}, \boxed{\texttt{[COMPILATION\_ERROR]}}\} \tag{12}$$

The specific reward values for topology validation and code execution errors are provided in Table 1. Additionally, the reward for $\boxed{\texttt{PASSED}}$ is 1.5, while no reward value is applied for successful YAML validation.

**Interaction Graph Complexity Reward Function**  To classify the interaction graph complexity according to difficulty levels, we define the function $\mathcal{S}_{\text{complex}}$ for the interaction topology graph density in Eq. 7. Given the task difficulty level $l$, each level is associated with a maximum allowed number of nodes $N_{\max}(l)$. For each turn $k$, the per-turn upper bound under the three difficulty levels is set to 4, 7, and 10, respectively.

$$N_{\max}^{(k)}(l) = \begin{cases} 4, & l = 1 \text{ (easy)}, \\ 7, & l = 2 \text{ (medium)}, \\ 10, & l = 3 \text{ (hard)}, \end{cases} \quad k \in \{1, 2\}. \tag{13}$$

If $|V|$ (the number of nodes, as defined in Eq. 4) exceeds this bound, the graph is considered overly complex and penalized accordingly. Finally, the overall interaction graph evaluation score is defined as

$$r_g(\mathcal{G}^{(k)}) = \begin{cases} \mathcal{S}_{\text{complex}}, & |V| \le N_{\max}(l), \\ \tanh\left(\frac{N_{\max}(l) - |V|}{N_{\max}(l)}\right), & \text{otherwise}. \end{cases} \tag{14}$$

## 3 EXPERIMENTS

### 3.1 EXPERIMENTAL SETUP

**Datasets and Metrics**  To comprehensively evaluate our approach in terms of performance, topology dynamics, and cost efficiency across problems of varying difficulty and type, we select two **basic code generation datasets** and three **contest-level code generation datasets**: **(1) Basic Code Generation Datasets:**  including HumanEval(Chen et al., 2021), MBPP(Austin et al., 2021); **(2) Contest-Level Code Generation Datasets:**  including APPS(Hendrycks et al., 2021), Live-CodeBench (V4)(Jain et al., 2024), and CodeContests(Li et al., 2022). The generated code is executed within a secure sandbox (Khan et al., 2023) environment. Model performance is then measured by the **pass@1** rate on each test set.

**Baselines**  To provide a comprehensive comparison and highlight the effectiveness of our approach, we evaluate against four categories of baselines: **(1)Vanilla:**  This setting reflects the capability of a single backbone model. We adopt `GPT-4o-mini` as the representative backbone. **(2)Classical Multi-Agent Systems:**  `AutoGen`(Wu et al., 2024), `MetaGPT`(Hong et al., 2024) and `MapCoder`(Islam et al., 2024). **(3)Multi-Agent Systems with Workflow Optimization:** `AFlow`(Zhang et al., 2024c), `FlowReasoner`(Gao et al., 2025) and `Chain-of-Agents`. **(4)Multi-Agent Systems with Topology Optimization:**  `GPTSwarm`(Zhuge et al., 2024), `AgentPrune`(Zhang et al., 2024a), `G-Designer` (Zhang et al., 2024b), and `MacNet`(Qian et al., 2024).(See Appendix B.1 for details.)

### 3.2 MAIN RESULTS

In this section, we provide extensive experimental evidence to analyze the effectiveness of our proposed **TopoWeaver-R1** method. Specifically, we evaluate its accuracy across diverse code generation tasks (Section 3.2.1), the dynamic adaptability of topology density and its superior cost-efficiency(Section 3.2.2), the fine-grained comparison across difficulty level(Section 3.2.3), and additional experimental results(Appendix C).

Table 2: Main performance of TopoWeaver-R1 on three competition-level and two basic code generation datasets *(mean ± std over 3 runs)*.

| Method | Contest-level Code Generation | | | | Basic Code Generation | | | Avg. |
|---|---|---|---|---|---|---|---|---|
| | APPS | LiveCodeBench | CodeContests | Avg. | HumanEval | MBPP | Avg. | |
| ***Vanilla*** | | | | | | | | |
| GPT-4o-mini | $20.3_{(\pm0.2)}$ | $26.3_{(\pm0.2)}$ | $18.6_{(\pm0.4)}$ | $21.7_{(\pm0.3)}$ | $87.6_{(\pm0.2)}$ | $73.5_{(\pm0.1)}$ | $80.5_{(\pm0.1)}$ | $51.1_{(\pm0.2)}$ |
| ***Classical Multi-Agent Systems (No Workflow/Topology Optimization)*** | | | | | | | | |
| AutoGen | $23.6_{(\pm2.3)}$ | $30.2_{(\pm1.5)}$ | $20.8_{(\pm1.9)}$ | $24.9_{(\pm1.9)}$ | $90.4_{(\pm0.8)}$ | $92.3_{(\pm0.4)}$ | $91.4_{(\pm0.6)}$ | $58.1_{(\pm1.3)}$ |
| MetaGPT | $\underline{51.3}_{(\pm1.4)}$ | $42.8_{(\pm1.3)}$ | $35.6_{(\pm1.2)}$ | $\underline{43.2}_{(\pm1.3)}$ | $95.8_{(\pm0.2)}$ | $92.3_{(\pm0.3)}$ | $94.1_{(\pm0.2)}$ | $\underline{68.7}_{(\pm0.6)}$ |
| MapCoder | $40.2_{(\pm0.9)}$ | $37.4_{(\pm1.1)}$ | $36.3_{(\pm0.7)}$ | $38.0_{(\pm0.9)}$ | $96.4_{(\pm0.5)}$ | $\underline{94.1}_{(\pm0.4)}$ | $95.3_{(\pm0.5)}$ | $66.6_{(\pm0.7)}$ |
| ***Multi-Agent Systems with Workflow Optimization*** | | | | | | | | |
| AFlow | $35.4_{(\pm1.7)}$ | $24.6_{(\pm1.1)}$ | $21.4_{(\pm1.5)}$ | $27.1_{(\pm1.4)}$ | $94.2_{(\pm0.3)}$ | $82.4_{(\pm0.1)}$ | $88.3_{(\pm0.2)}$ | $57.7_{(\pm0.8)}$ |
| FlowReasoner | $39.1_{(\pm1.9)}$ | $43.8_{(\pm2.1)}$ | $\underline{37.7}_{(\pm1.6)}$ | $40.2_{(\pm1.9)}$ | $\underline{97.3}_{(\pm0.5)}$ | $93.9_{(\pm0.7)}$ | $\underline{95.6}_{(\pm0.6)}$ | $67.5_{(\pm1.3)}$ |
| Chain-of-Agents(32B) | $41.6_{(\pm1.3)}$ | $\underline{44.9}_{(\pm1.2)}$ | $34.6_{(\pm1.2)}$ | $40.3_{(\pm1.2)}$ | $95.3_{(\pm0.2)}$ | $90.2_{(\pm0.3)}$ | $92.8_{(\pm0.2)}$ | $67.9_{(\pm0.6)}$ |
| ***Multi-Agent Systems with Topology Optimization*** | | | | | | | | |
| GPTSwarm | $36.5_{(\pm2.1)}$ | $40.8_{(\pm2.5)}$ | $31.6_{(\pm3.0)}$ | $36.3_{(\pm2.5)}$ | $94.8_{(\pm1.1)}$ | $91.6_{(\pm1.3)}$ | $93.2_{(\pm1.2)}$ | $64.8_{(\pm1.9)}$ |
| AgentPrune(Complex) | $38.6_{(\pm1.9)}$ | $41.7_{(\pm2.1)}$ | $33.5_{(\pm0.8)}$ | $37.9_{(\pm1.6)}$ | $96.1_{(\pm0.5)}$ | $91.8_{(\pm0.6)}$ | $94.0_{(\pm0.7)}$ | $65.9_{(\pm1.1)}$ |
| AgentPrune(Layered) | $39.3_{(\pm1.6)}$ | $41.9_{(\pm1.8)}$ | $31.4_{(\pm0.9)}$ | $37.5_{(\pm1.4)}$ | $96.6_{(\pm0.7)}$ | $92.3_{(\pm0.3)}$ | $94.5_{(\pm0.5)}$ | $66.0_{(\pm1.0)}$ |
| MacNet(Complex) | $37.6_{(\pm0.8)}$ | $39.4_{(\pm0.7)}$ | $28.7_{(\pm0.7)}$ | $35.2_{(\pm0.7)}$ | $95.8_{(\pm0.4)}$ | $89.4_{(\pm0.2)}$ | $92.6_{(\pm0.3)}$ | $63.9_{(\pm0.5)}$ |
| MacNet(Layered) | $36.9_{(\pm0.6)}$ | $40.3_{(\pm0.5)}$ | $28.9_{(\pm0.8)}$ | $35.4_{(\pm0.6)}$ | $95.2_{(\pm0.2)}$ | $90.3_{(\pm0.3)}$ | $92.8_{(\pm0.3)}$ | $64.1_{(\pm0.5)}$ |
| G-Designer | $37.2_{(\pm1.5)}$ | $38.8_{(\pm1.3)}$ | $26.9_{(\pm1.2)}$ | $34.3_{(\pm1.3)}$ | $95.6_{(\pm0.9)}$ | $90.9_{(\pm0.8)}$ | $93.2_{(\pm0.9)}$ | $63.7_{(\pm1.1)}$ |
| TopoWeaver-R1(3B) | $\mathbf{58.8}_{(\pm0.3)}$ | $\mathbf{46.3}_{(\pm0.4)}$ | $\mathbf{38.8}_{(\pm0.5)}$ | $\mathbf{48.0}_{(\pm0.3)}$ | $\mathbf{97.5}_{(\pm0.1)}$ | $\mathbf{95.1}_{(\pm0.2)}$ | $\mathbf{96.3}_{(\pm0.2)}$ | $\mathbf{72.1}_{(\pm0.3)}$ |

### 3.2.1 CODE GENERATION PERFORMANCE

As shown in Table 2, our approach consistently achieves the highest accuracy across all five datasets. In the contest-level benchmarks, **TopoWeaver-R1** reaches pass@1 accuracies of 58.8%, 46.3%, and 38.8% on APPS, LiveCodeBench (v4), and CodeContests, respectively, **outperforming the second-best methods by absolute margins of 14.6%, 3.1%, and 1.1% percentage points**. In the basic code generation tasks, our method achieves pass@1 accuracies of 97.5% on HumanEval and 95.1% on MBPP, **surpassing the second-best methods by absolute margins of 1.0% and 0.7% percentage points, respectively**(See Appendix C.1 for details).

### 3.2.2 COMPARISON OF DYNAMIC TOPOLOGY GENERATION AND COST EFFICIENCY

Table 3: APPS results comparing TopoWeaver-R1 with baselines on performance, cost, and average topology density.

| Dataset | Method | Performance | Prompt Tokens | Completion Tokens | $\mathcal{S}_{\text{complex}}$ (↑) |
|---|---|---|---|---|---|
| APPS | AFlow | 35.4 | 531450 | 184800 | 3.7 |
| | FlowReasoner | 39.1 | 437250 | 148050 | 2.4 |
| | Chain-of-Agents (32B) | $\underline{41.6}$ | $\underline{334650}$ | $\underline{134250}$ | $\underline{4.1}$ |
| | GPTSwarm | 36.5 | 381450 | 155400 | 3.5 |
| | AgentPrune (Layered) | 39.3 | 364950 | 141150 | 3.8 |
| | MacNet (Layered) | 36.9 | 472950 | 200100 | 2.9 |
| | G-Designer | 37.2 | 320550 | 139200 | 3.6 |
| | **TopoWeaver-R1 (3B)** | **58.8** | **277600** | **79800** | **5.2** |

In Table 3 and Figure. 1(b), using the APPS dataset as a case study, we visually compare our approach with six alternative workflow and topology optimization methods to assess both **cost efficiency and average topology density**. For cost, we report the consumption of **Prompt Tokens** and **Completion Tokens**; for density, we adopt the average score $\mathcal{S}_{\text{complex}}$ from Eq. 7, where larger values indicate lower (sparser) topology density. The table shows that **TopoWeaver-R1** attains the lowest consumption of prompt tokens and the consumption of completion tokens and the highest average $\mathcal{S}_{\text{complex}}$ (i.e. the sparsest interaction topology), while still achieving the best accuracy. This

indicates that, in contest-level code generation, our method delivers higher performance at lower cost.

### 3.2.3 AVERAGE TOPOLOGY DENSITY COMPARISON BY DIFFICULTY LEVEL

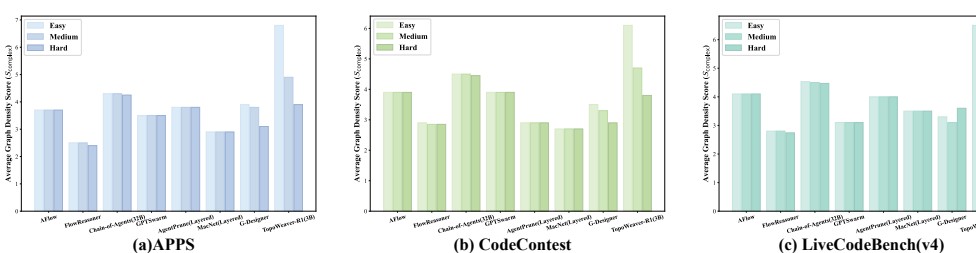

(a)APPS    (b) CodeContest    (c) LiveCodeBench(v4)

Figure 3: Comparison of the average topology density ($\mathcal{S}_{complex}\uparrow$ sparser) across three competition-level code datasets at three difficulty levels.

Moreover, Figure 3 presents a fine-grained comparison across difficulty levels on three contest-Level datasets . Our method modulates topology density with problem difficulty. It uses sparser graphs for easier instances and denser graphs for harder ones, thereby reducing token cost on easy cases while preserving accuracy on hard cases. In contrast, competing methods exhibit little or no density adaptation across difficulty, which leads to unnecessary token expenditure.

### 3.3 ABLATION STUDY

**Impact of Supervised Fine-tuning and Reinforcement Learning**    We examine whether CoT-based SFT is necessary by comparing (i) direct RL without SFT and (ii) SFT followed by RL. We report three metrics to make the performance factors explicit: (1) **Performance**, measured by code-generation `pass@1`; (2) $\mathcal{S}_{complex}$ for graph density; and (3) **Valid topology (%)**, the percentage of topologies that satisfy the formatting constraints and the difficulty-specific density cap. From Table 4, the SFT stage is crucial for producing valid and executable topologies: small open-source backbones trained without SFT rarely meet the required format and density, and consequently fail to produce correct code. In contrast, SFT only (without RL) attains a moderate valid-topology rate;

Table 4: Ablation study on Training Strategies and Reward Design.

| | Method | APPS | | | HumanEval | | |
|---|---|---|---|---|---|---|---|
| | | Performance | $\mathcal{S}_{complex}$ ($\uparrow$) | Valid Topo (%) | Performance | $\mathcal{S}_{complex}$ ($\uparrow$) | Valid Topo (%) |
| | Full Model | **58.8** | **5.2** | **100** | **97.5** | **5.8** | **100** |
| Training Strategies | *w/o* SFT | – | – | 15 | – | – | 13 |
| | *w/o* RL | 29.8 | 2.7 | 56.5 | 90.2 | 3.2 | 57.2 |
| Reward | *w/o* $r_e(\mathcal{E}_{yaml\_errors})$ | 30.3 | 2.9 | 56.8 | 91.4 | 3.0 | 58.1 |
| | *w/o* $r_e(\mathcal{E}_{code\_errors})$ | 35.5 | 5.0 | 96.4 | 93.1 | 5.6 | 99.2 |
| | *w/o* $S_{node}$ | 49.2 | 3.8 | 85.8 | 96.9 | 4.8 | 87.2 |
| | *w/o* $S_{edge}$ | 45.5 | 4.5 | 89.3 | 96.1 | 4.6 | 90.5 |
| | *w/o* $S_{diameter}$ | 48.3 | 3.9 | 91.7 | 95.3 | 4.1 | 93.4 |
| | *w/o* $r_g(\mathcal{G}^{(k)})$ | 52.6 | 3.0 | 83.2 | 97.2 | 3.4 | 85.6 |

**Impact of Multi-objective Reward Design**    Table 4 summarizes the impact of individual reward components on model performance. We observe that the YAML-format error term $r_e(\mathcal{E}_{yaml\_errors})$ has the strongest effect on the valid-topology rate, whereas the code-execution error term $r_e(\mathcal{E}_{code\_errors})$ most strongly affects code accuracy (pass@1). The three topology-density sub-rewards $S_{node}$, $S_{edge}$, and $S_{diameter}$ influence both density control and accuracy to different extents, with *w/o* $S_{node}$ causing the largest degradation in code-generation performance. Lower topology density (especially without $r_g(\mathcal{G}^{(k)})$) can reduce accuracy by limiting agents and interactions. With the full reward, optimizing density and accuracy together guides the policy to suitable interaction patterns and densities, boosting performance while keeping token usage efficient.

## 4 RELATED WORKS

### 4.1 LLM-BASED MAS FOR CODE GENERATION

LLM-based multi-agent systems have shown promise in code generation(Huang et al., 2023; Nunez et al., 2024; Ishibashi & Nishimura, 2024). Frameworks such as MetaGPT(Hong et al., 2024) and AutoGen(Wu et al., 2024) introduce software development workflows and role-playing to enhance collaboration. These approaches, however, face challenges in competition-level settings, which demand deeper algorithmic reasoning and precise implementation. MapCoder(Islam et al., 2024) using multi-round planning, retrieval scoring, and algorithmic tutorials to achieve notable results. Still, since competition problems vary widely in difficulty, fixed agent frameworks often incur unnecessary overhead—such as redundant interaction and roles—on simpler tasks, motivating more adaptive solutions.

### 4.2 TOPOLOGY OPTIMIZATION AND GENERATION FOR MAS

Recent works (Zhuge et al., 2024; Zhang et al., 2024c) have explored optimizing interaction topologies in multi-agent systems to improve efficiency. Graph pruning methods, such as AgentPrune (Zhang et al., 2024a) and AgentDropout(Wang et al., 2025a), iteratively reduce interaction graphs to a minimal structure. However, these rely on a fixed topology per task. Dynamic orchestration methods(Zhang et al., 2025; Dang et al., 2025) select a topology through multi-round optimization but still finalize it before execution. Generation-based approaches like G-Designer(Zhang et al., 2024b) produce a topology from problem descriptions, allowing finer adaptation but remaining static thereafter. A common limitation is the tendency to converge to uniformly sparse structures, lacking fine-grained difficulty awareness.

Agentic reinforcement learning (RL) methods(Wang et al., 2025b; Jin et al., 2025) have recently introduced new paradigms for large language models, enabling them to move beyond single-turn outputs toward multi-turn interactions with the environment and tool usage. These approaches optimize the model by incorporating external tools or agent–environment interactions into the agent's output as part of a complete trajectory, thereby endowing the agent with the capability of multi-round interaction with its environment. Inspired by this line of work, several studies have further explored end-to-end optimization of agent workflows by leveraging full interaction trajectories, as seen in FlowReasoner(Gao et al., 2025) and Chain-of-Agents(Li et al., 2025). *While FlowReasoner introduces local parallelism within certain operator blocks, it still cannot express rich graph-structured interactions; Chain-of-Agents, in contrast, follows a purely sequential workflow without any parallel branches.* Departing from these lines, we propose an Agentic RL-based approach centered on a central orchestrator that dynamically generates and iteratively refines interaction topologies in natural language, conditioned on execution feedback. A key innovation is a difficulty-aware density reward, which explicitly modulates topology sparsity according to problem difficulty.

## 5 CONCLUSION

In summary, TopoWeaver-R1 establishes a new paradigm for competition-level code generation by integrating difficulty-aware reinforcement learning with multi-turn topology evolution. By training an orchestrator agent to dynamically generate and refine interaction topologies through execution feedback and density-aware rewards, our method achieves fine-grained adaptability across problem difficulties. This paradigm advances multi-agent code generation toward systems that are not only accurate, but also cost-efficient and scalable.

ETHICS STATEMENT

This work complies with the ICLR Code of Ethics. All datasets used are publicly available, and no human subjects or sensitive personal data were involved. The research is conducted solely for scientific purposes, with no foreseeable risks of harmful use or conflicts of interest.

REPRODUCIBILITY STATEMENT

We have taken several steps to ensure the reproducibility of our work. The paper provides detailed descriptions of the proposed method (Section.2), training settings (Appendix.B.2), evaluation protocols (Paragraph.3.1), and ablation studies (Section.3.3). All datasets (Paragraph.3.1) used are publicly available, and we describe the data preprocessing steps in the supplementary materials. Pseudocode and proofs of the theoretical results are included in the appendix. We will also upload a compressed package containing the complete main code for reproduction.

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

## A  LLM Usage Statement

We used large language models (LLMs) solely as general-purpose assistive tools. Specifically, LLMs were employed for language polishing, grammar refinement, and improving the clarity of the manuscript. In addition, we occasionally used LLMs to assist in debugging minor programming issues (e.g., syntax errors or code formatting), but not for problem solving, experimental design, data analysis, or ideation of the research. The substantive contributions—including research ideas, methodology design, implementation, analysis, and writing of technical content—were entirely the work of the authors.

## B  Supplementary Experimental Setup

### B.1  Supplementary Details on Baselines

To provide a comprehensive comparison and highlight the effectiveness of our approach, we evaluate against four categories of baselines: **(1)Vanilla:** This setting reflects the capability of a single backbone model. We adopt `GPT-4o-mini` as the representative backbone. **(2)Classical Multi-Agent Systems:** This category includes three representative frameworks: `AutoGen`(Wu et al., 2024) is a general-purpose multi-agent framework, `MetaGPT`(Hong et al., 2024) is designed for generic coding tasks, and `MapCoder`(Islam et al., 2024)targets competitive programming code generation. **(3)Multi-Agent Systems with Workflow Optimization:** This category comprises three systems: `AFlow`(Zhang et al., 2024c) leverages search-based methods to optimize the workflow, while `FlowReasoner`(Gao et al., 2025) and `Chain-of-Agents` are recent reinforcement learning approaches that optimize multi-agent workflows end-to-end. **(4)Multi-Agent Systems with Topology Optimization.** This category covers `GPTSwarm`(Zhuge et al., 2024), `AgentPrune`(Zhang et al., 2024a), `G-Designer` (Zhang et al., 2024b), and `MacNet`(Qian et al., 2024). These approaches explicitly focus on optimizing the agent interaction topology.

For multi-agent baselines, **we align the role definitions and system prompts with those used in our method**. For workflow and topology optimization methods, we set the maximum number of participating agent nodes to **20**. This matches the upper bound of topology density in our framework when solving the most challenging problems with up to two interaction turns, ensuring a fair comparison. Following the setup in MacNet, we note that our topology can be viewed as an evolved variant of layered graphs. Our topology exhibits an intermediate density, between complex and layered graphs. **To ensure comprehensive and reliable evaluation, we therefore compare *AgentPrune* and *MacNet* under both complex-graph and layered-graph initialization settings.**

### B.2  Implementation Details

For TopoWeaver-R1, we use Qwen2.5-3B-Instruct (Yang et al., 2024) as the backbone. During the SFT stage, we adopt the LLaMA-Factory framework (Zheng et al., 2024) for training. Specifically, we utilize 4500 synthetic samples constructed from three contest-level code generation datasets across three difficulty levels (see Section 2.2 for details). The training is performed with an initial learning rate of $1 \times 10^{-4}$, a batch size of 4, and LoRA-based fine-tuning, while all other hyperparameters are kept at their default values. During the reinforcement learning stage, we implement GRPO using the Verl (Sheng et al., 2025) framework with vLLM for generation(code development based on Search-R1 (Jin et al., 2025)). We set the group size to $G = 8$, with a batch size of 8, a learning rate of $1 \times 10^{-6}$, a policy temperature of 1, and a maximum completion length of 4096 tokens. To balance performance and computational cost, we further limit the maximum number of turns (i.e., multi-agent interaction turns) to 2. Throughout training, individual agents are executed with `gpt-4.1-nano` and interact in real time with a code execution sandbox to obtain authentic runtime feedback. Both stages are conducted on a 4-GPU A800 cluster.

### B.3  Progressive Quality Filtering for SFT Data

*Our training data consist of valid, executable, and semantically correct topologies generated by GPT-4O-mini under code-oriented tasks. All data are produced using the same role configuration and topology density constraints adopted in our orchestrator. The second-turn interaction topologies*

*are real and valid structures obtained from actual error messages and historical multi-agent logs, rather than synthetic approximations.*

*We first perform strict YAML syntax verification to ensure that each example is well-formed and can be parsed by standard YAML loaders. This step guarantees that all topologies can be safely converted into JSON objects for subsequent processing, preventing malformed or incomplete structures from entering the dataset. Second, we apply semantic validation using a predefined JSON_SCHEMA. After converting each YAML topology into JSON, we verify that it satisfies all orchestration constraints. The validation rules include: (1) The ref field of all agents in the first timestep must be empty. (2) For every agent, all agent IDs listed in its ref field must correspond to agents that have appeared in earlier timesteps. These schema-level checks ensure the structural consistency and logical correctness of the generated topologies. We further remove duplicate topologies and preserve only those that successfully interact with the execution environment. This step ensures that the topologies are not merely syntactically valid but are also actionable and executable within the orchestrator runtime. All remaining samples are re-validated using GPT-4o-mini to ensure semantic soundness, consistency, and correctness. Finally, we manually inspect a randomly sampled 5% subset of the data to further confirm high-quality labeling and structural validity.*

## B.4 System Prompt for Orchestrator Agent

You are a Orchestrator agent. Your goal is to coordinate a multi-agent team to solve the given code problem by generating a YAML-formatted interaction plan.
Each plan should specify:
- Which agents to activate at each step;
- Which previous agents' outputs are referenced.
Agent types:
- `<planner>`: plans algorithmic strategy.
- `<searcher>`: retrieves relevant knowledge.
- `<algorithmer>`: analyzes problem structure and decomposes it into key algorithmic components or subroutines.
- `<coder>`: generates code based on other agents' information.
- `<debugger>`: fixes incorrect code (only used after a `<coder>`).
- `<tester>`: verifies code (must be used in the last step, referencing `<coder>` or `<debugger>`)
Format:
Output only the **YAML** plan.
Each step includes one or more agents with optional references.
---
### Notes:
1. There are three levels of difficulty, arranged from low to high as follows: introductory, interview, competition.
2. Determine whether the task difficulty is introductory, interview, or competition.
3. **Dynamically adjust the number of steps and agents** based on the difficulty of the problem.
4. For **more difficult problems**, **involve more agents** if necessary. \
For **simpler problems**, you may **reduce both the number of agents** and **the number of steps** involved.
5. The last step must include a `<tester>` referencing at least one of `<coder>` or `<debugger>`
6. Execute up to max **{max_turn_num}** rounds in total, until the code passes verification by the `<tester>`.
7. In the first step, all agents must have empty `ref` fields.
---
### The Code Problem is:
**Task**: `{question}`
---
### Your output should be a **YAML-formatted** plan only.
your output: \n

Figure 4: The figure shows the system prompt for the orchestrator agent.

We show in the figure the system prompt of the trained orchestrator agent.

## C Additional Experimental Results

### C.1 Code Generation Performance Analysis

We observe that **MetaGPT**, a code-oriented multi-agent framework with a fixed interaction scheme, achieves the second-best performance on average. Among optimization-oriented approaches, the two end-to-end reinforcement learning methods, **FlowReasoner** and **Chain-of-Agents**, rank next and narrowly trail **MetaGPT** in average results. By contrast, topology optimization methods under-

perform, likely because their learned topologies remain comparatively rigid and struggle to adapt to the highly variable and complex nature of competitive programming tasks. **G-Designer** is a method that generates interaction graphs based on the given problem. However, we observe that although these methods are adapted to different tasks, the difficulty of competition-level problems is hard to distinguish intuitively, and thus such adaptations do not lead to significant improvements in code performance. Within this family, **AgentPrune** and **MacNet** perform better under layered-graph initialization, suggesting that for relatively sequential code-generation tasks, layered graphs provide a more suitable inductive bias than unstructured complex graphs. **Building on this, TopoWeaver-R1 retains the inductive bias of layered graphs yet adapts dynamically per problem, yielding state-of-the-art overall accuracy.**

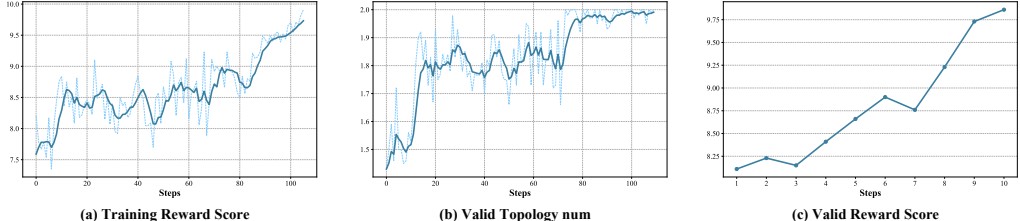

(a) Training Reward Score  (b) Valid Topology num  (c) Valid Reward Score

Figure 5: The figure shows the dynamics of three key metrics during RL training: (a) training reward, (b) average number of valid two-turn topologies, and (c) validation reward. The results indicate that our method progressively converges toward generating topologies with reasonable density and achieving accurate code problem solving in later training stages.

## C.2   ANALYSIS ON THE RL TRAINING CURVE

To better understand the training dynamics of the reinforcement learning stage, we plot the trajectories of (i) the average reward, (ii) the count of topologies passing the density check, and (iii) the validation score over the first 110 RL training steps (Figure 6). Our key observations are as follows: all three metrics increase steadily with training, indicating that the self-critic RL procedure is stable and makes consistent progress. These results further demonstrate that our method trains effectively and remains stable.

## C.3 Case Study

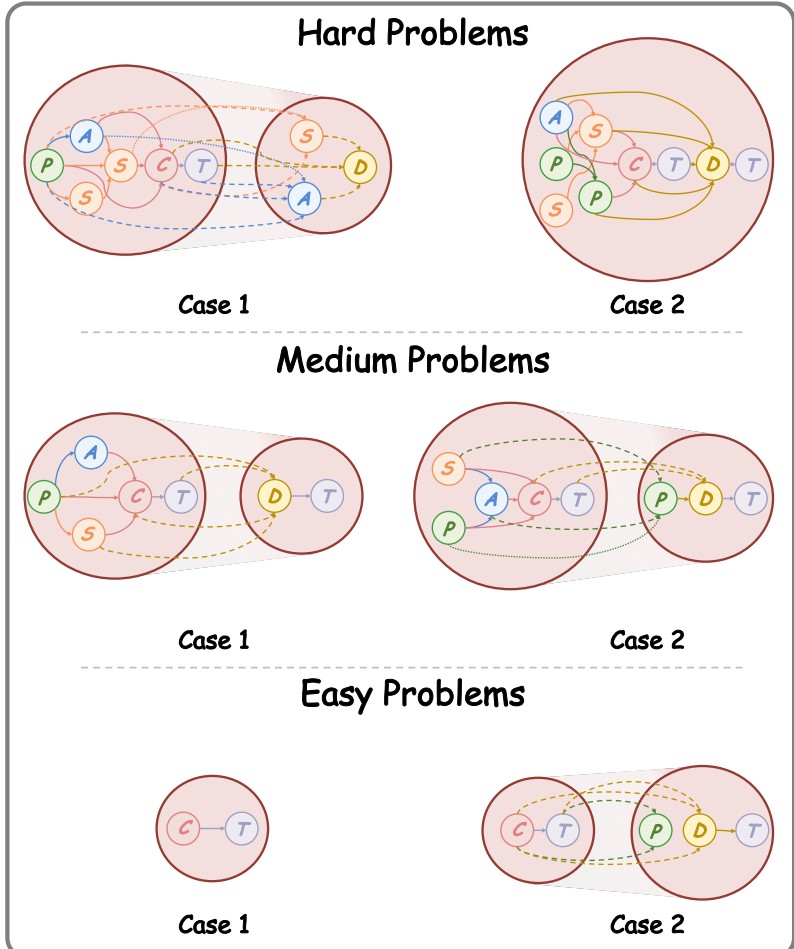

Figure 6: The figure shows the generated interaction topologies for two problem cases at each difficulty level.

Based on the generated cases shown in the figure, our method exhibits the following characteristics. First, it can generate different initial interaction topologies tailored to the characteristics of individual problems, with topology density varying according to difficulty. Second, the method dynamically adjusts the second-round topology based on the execution results of the first round; this adjustment does not necessarily reduce the number of agents, as additional agents may be introduced when errors occur. Finally, when agents from the first round reappear in the second round, their behavior evolves according to their prior outputs, thereby achieving iterative evolution. These characteristics highlight the customizability and adaptability of our approach, which in turn enhance system performance while reducing costs in a fine-grained manner.

## C.4 Zero-Shot Transfer to Unseen Roles and Task Types

*To evaluate the transferability of our orchestrator to unseen problem types and newly introduced agent roles, we conducted a small-scale study on 50 filtered samples from the GAIA (Mialon et al., 2023) dataset. These samples were strictly restricted to tasks where the inputs consist solely of single-modality textual descriptions, which differ substantially from the code-generation domain used for training.*

*No additional training was performed. Instead, we expanded the orchestrator's role pool by adding two previously unseen roles: an online search agent* `<online_searcher>` *and a visual validation agent* `<visual_checker>`, *together with their corresponding tool interfaces. Using the*

*original trained model, the orchestrator was able to naturally integrate these new roles into the generated interaction topologies, despite never encountering them during SFT or RL training.*

*Under this strict zero-shot transfer setting, the framework achieved a success rate of **15.8%** on the selected GAIA samples, demonstrating that the orchestrator exhibits non-trivial generalization to unseen domains, unseen task types, and unseen agent capabilities.*

## C.5 SUPPLEMENTARY CROSS-DOMAIN EXPERIMENTS

*While our method was initially designed with a focus on competition-level code generation, this focus was a deliberate choice rather than a limitation. Competition-level tasks provide a highly challenging and well-instrumented testbed that allows us to rigorously examine dynamic topology evolution under strict execution feedback, token constraints, and difficulty-aware limits. Aligned with our research interests, our goal was to develop a specialized multi-agent orchestration algorithm for this domain, offering a complementary perspective to prior multi-agent architecture studies that emphasize broad task coverage. Nevertheless, our method is inherently generalizable. To address the reviewer's concern, we additionally evaluate the cross-domain applicability of our approach.*

*Following the role definitions and data filtering strategy used in Chain-of-Agents **?**, we expanded the agent role pool in our orchestrator's system prompt. The newly introduced roles include: `<online_searcher>` for web-based retrieval, `<thinker>` for complex reasoning, `<verifier>` for answer verification, and `<planner>` for task decomposition and high-level orchestration. All roles were redefined and implemented for reasoning-centric tasks. We selected subsets from three representative datasets—**GAIA(Mialon et al., 2023)**, **HLE(Phan et al., 2025)**, and **PopQA(Mallen et al., 2023)**—to evaluate multi-hop reasoning and question answering.*

*For the reward function, we retain the YAML validation and topology-density components, which remain general across tasks and domains. To adapt the pipeline, we replace the code-execution validator with an LLM-based answer validator and simplify the reward to a binary scheme: 1 for correctness and 0 otherwise. All other training and inference settings remain unchanged. We retrained our model under this configuration and report results below.*

Table 5: Cross-domain evaluation of TopoWeaver-R1 on GAIA, HLE, and PopQA. Results are reported as mean $\pm$ std over three seeds.

| Method | Backbone | GAIA L1 | GAIA L2 | GAIA L3 | GAIA Avg. | HLE Avg. | PopQA |
|---|---|---|---|---|---|---|---|
| Chain-of-Agents | 7B | $69.2_{(\pm 0.8)}$ | $50.9_{(\pm 0.7)}$ | $33.3_{(\pm 1.1)}$ | $50.8_{(\pm 0.8)}$ | $18.0_{(\pm 0.6)}$ | $46.5_{(\pm 1.3)}$ |
| **TopoWeaver-R1 (ours)** | **3B** | $72.0_{(\pm 0.4)}$ | $53.4_{(\pm 0.3)}$ | $36.1_{(\pm 0.5)}$ | $53.8_{(\pm 0.4)}$ | $22.6_{(\pm 0.2)}$ | $50.3_{(\pm 0.3)}$ |

*The results demonstrate that TopoWeaver-R1 outperforms Chain-of-Agents across all datasets despite using a considerably smaller backbone (3B vs. 7B). Our method achieves strong accuracy and maintains low variance across seeds, highlighting both the robustness and adaptability of the proposed topology optimization framework. These findings provide further evidence that our approach generalizes beyond code generation and can be transferred to new reasoning-oriented domains with minimal modification.*

## D DETAILED DEFINITIONS OF TOPOLOGY NOTIONS

**Agent Node Notations**    Each agent node $v_i^{(k)}$ is defined as:

$$v_i^{(k)} = \left\{ \mathsf{Type}_i,\ \mathsf{Base}_i,\ \mathsf{Role}_i^{(k)},\ \mathsf{View}_i^{(k-1)},\ \mathsf{Mem}_i^{(<k)} \right\} \tag{1}$$

The $\mathsf{Type}_i$ field specifies one of three agent categories: (1) The Orchestrator agent is a locally deployed large language model (LLM) proposed and trained in this work, designed to generate multi-turn YAML interaction topologies in an end-to-end orchestrator and to manage the execution of multiple agents; (2) The LLM-agent is a prompt-conditioned LLM (open-source or via API)

that is assigned a role; and (3) the ToolAgent, which is equipped with callable external APIs such as retrieval engines or code execution tool. $\mathsf{Role}_i^{(k)}$ is the turn-specific role/prompt (e.g., `<planner>`, `<coder>`). $\mathsf{View}_i^{(k-1)}$ is the orchestrator-curated visible context for this agent, including selected outputs from its dependencies and possibly from last turn. Finally, $\mathsf{Mem}_i^{(<k)}$ stores the cross-turn history of agent $i$ prior to turn $k$.

**Notations for Agent Communication Edges**  In our framework, the edge set is constructed directly from the `ref` fields specified in the YAML plan, and we categorize edges into three types. First, *intra-turn edges* $\mathcal{E}^{\text{intra}} \subseteq \mathcal{V}^{(k)} \times \mathcal{V}^{(k)}$ connect agents within the same turn according to their declared references. Second, *inter-turn cross-agent edges* $\mathcal{E}^{\text{cross}} \subseteq \mathcal{V}^{(k-1)} \times \mathcal{V}^{(k)}$ capture dependencies across two consecutive turns when an agent in turn $t$ explicitly references outputs from other agents in turn $k-1$. Third, *inter-turn self-edges* $\mathcal{E}^{\text{self}} \subseteq \{(v_i^{(k-1)}, v_i^{(k)}) \mid v_i \in \mathcal{V}\}$ are automatically added whenever the same agent is invoked across two consecutive turns, allowing it to incorporate and refine its own previous outputs.

**Orchestrator-Guided Multi-Agent Interaction.**  Given a task $x$, the orchestrator agent emits a YAML plan for turn $k$. The plan tokens are sampled from the orchestrator policy and deterministically decoded into a strict layered DAG $\mathcal{G}^{(k)}$ (see Eq. 2). The node set $\mathcal{V}^{(k)}$ is instantiated with *LLM-agents* and *ToolAgents*; execution follows the step (layer) order implied by $\mathcal{G}^{(k)}$: agents within the same step run in parallel, and there are no intra-step edges. We intentionally exclude intra-step interaction to facilitate parallel execution and reduce scheduling complexity. Although a fully connected DAG allows richer expressiveness, we find that enforcing structural sparsity within steps improves interpretability, efficiency, and learning stability. For a node $v_i^{(k)}$, the turn-$k$ output is produced as

$$M_i^{(k)} \sim \mathcal{P}_{\theta_i}\big(M \mid x, \mathsf{Role}_i^{(k)}, \mathsf{View}_i^{(k-1)}, \mathsf{Mem}_i^{(<k)}, \{M_j^{(k)} : (v_j^{(k)}, v_i^{(k)}) \in \mathcal{E}^{(k)}\}\big). \qquad (15)$$

$\mathcal{E}^{(k)}$ is the intra-turn dependency set (a strict layered DAG) parsed from the YAML `ref` fields; $\{M_j^{(k)} : (v_j^{(k)}, v_i^{(k)}) \in \mathcal{E}^{(k)}\}$ collects the outputs of all in-neighbors of $v_i^{(k)}$ in turn $k$; $\mathsf{Role}_i^{(k)}$ is the turn-specific role/prompt of $v_i$; $\mathsf{View}_i^{(k-1)}$ is the orchestrator-curated summary of the previous turn (topology/error cues) provided as read-only context; $\mathsf{Mem}_i^{(<k)}$ is the agent-local cross-turn memory prior to turn $k$; $\mathcal{P}_{\theta_i}$ denotes the agent's conditional kernel (LLM likelihood for language agents; deterministic operator such as retriever $r$ or executor $\xi$ for ToolAgents); and $M_i^{(k)}$ is the outputs produced by $v_i^{(k)}$ in turn $k$.

After execution, each agent appends its output to its memory, $\mathsf{Mem}_i^{(\leq k)} = \bigcup_{t=1}^{k} \{M_i^{(t)}\}$. Each turn concludes with a tester agent that executes the candidate code and returns a status $s^{(k)}$, which can either be `PASSED` or one of the errors from the set $\mathcal{E}_{\text{errors}}$ defined in Eq.12. If $s^{(k)} = $ `PASSED`, the process stops and the solution is accepted. Otherwise, the orchestrator agent collects the observation $\mathcal{O}^{(k)} = \{\mathcal{E}_{\text{errors}}, \mathcal{L}_{\text{logs}}, \mathcal{G}^{(k)}\}$, which includes error types $\mathcal{E}_{\text{errors}}$, execution logs $\mathcal{L}_{\text{logs}}$, and the turn-$k$ topology trace $\mathcal{G}^{(k)}$. Based on the observation, the orchestrator agent generates the next-turn interaction graph via Eq.1 and Eq.2. During this process, the orchestrator decides which agents to *reuse* from memory, which to *rerun*, and which to *activate*. The orchestrator continues to regenerate the topology for each turn as needed until the code result is `PASSED` or the maximum number of turns $K$ is reached.

**Definition 1.** For a strict layered DAG $\mathcal{G}^{(k)}$, the node set $\mathcal{V}^{(k)}$ is divided into $b$ independent sets $\{\mathcal{V}_1^{(k)}, \dots, \mathcal{V}_b^{(k)}\}$ with a well-defined layer structure. It has the following properties:

(**Sequentiality**) for any edges $(u, v)$, it satisfies that $u \in \mathcal{V}_i^{(k)}$, $v \in \mathcal{V}_j^{(k)}$, and $i < j$.

(**Conciseness**) for any nodes $u \in \mathcal{V}_i^{(k)}$ where $i \neq b$, there must exist an edge $(u, v)$ such that $v \in \mathcal{V}_j^{(k)}$, where $i < j$.

---

**Algorithm 1** Online Topology Generation Workflow of TopoWeaver-R1

---

**Require:** Input query $x$, Policy model $\pi_\theta$, Maximum Rounds $K$
**Ensure:** Final output $z$
 1: initialize history $H$.
 2: initialize local memory $\{\text{Mem}_i\}$ for each agent $v_i$
 3: initialize $z \leftarrow \varnothing$
 4: **for** round $k \leftarrow 1$ to $K$ **do**
 5:     $o_k = (o_{k,1}, \ldots, o_{k,|o_k|}) \sim \pi_\theta(\cdot | x, H_k)$
 6:     **if** no valid YAML detected in $o_k$ **then**
 7:         $y_k \leftarrow \text{YAMLCheck}(o_k)$
 8:         $H_{k+1} \leftarrow H_k.\text{append}((o_k, y_k))$
 9:         continue
10:     **end if**
11:     $\mathcal{G}^{(k)} = \text{DecodeTopo}(o_k)$
12:     $z_k = (z_k^{\text{roles}}, z_k^{\text{code}}) \leftarrow \text{ExecRun}(x, \mathcal{G}^{(k)}, H_k)$
13:     **if** $\boxed{\texttt{PASSED}}$ in $z_k^{\text{code}}$ **then**
14:         break                                                                                  ▷ Early stopping
15:     **end if**
16:     $H_{k+1} \leftarrow H_k.\text{append}((\mathcal{G}^{(k)}, z_k))$
17:     $z \leftarrow z + z_k$
18: **end for**
19: **return** final output $z$
20: **procedure** EXECRUN$(x, \mathcal{G}^{(k)}, H_k)$
21:     initialize $z_k^{roles} \leftarrow \varnothing$
22:     **for** layer in $\mathcal{G}^{(k)}$ **do**
23:         Run $\{v_i \mid v_i \in \text{layer}\}$ in parallel:
24:             $M_i^{(k)} \sim \mathcal{P}_{\theta_i}\big(M \mid x, \text{Role}_i^{(k)}, \text{View}_i^{(k-1)}, \text{Mem}_i^{(<k)}, \{M_j^{(k)} : (v_j^{(k)}, v_i^{(k)}) \in \mathcal{E}^{(k)}\}\big)$
25:             Add $M_i^{(k)}$ to $\text{Mem}_i$
26:             $z_k^{roles} \leftarrow z_k^{roles} + M_i^{(k)}$
27:     **end for**
28:     Extract code $\text{code}_k$ from $z_k^{roles}$
29:     $z_k^{\text{code}} \leftarrow \text{tester}(\text{code}_k)$
30: **return** $(z_k^{\text{roles}}, z_k^{\text{code}})$
31: **end procedure**

---

## D.1    ALGORITHM WORKFLOW OF TOPOWEAVER-R1

We conclude the overall algorithm workflow of TopoWeaver-R1 in Algorithm 1

## D.2    THEORETICAL DERIVATION AND PROOF OF TOPOLOGY DENSITY

**From Token Cost to Topology Density**    In order to achieve the goal of cost saving, we define the topology density based on the cost efficiency. Now we give the mathematical derivation here to show that in MAS, the complexity of agent interactions can be formally mapped into graph properties to quantify operational costs.

We first model the interaction per round as a graph $\mathcal{G}^{(k)} = (\mathcal{V}^{(k)}, \mathcal{E}^{(k)})$, where vertices $\mathcal{V}^{(k)}$ represent agents and edges $\mathcal{E}^{(k)}$ capture dependency relationships in round $k$.

To eliminate the influence of difficulty on topology scale, we prefer the average cost on each agent. For each agent, the token cost mainly consists of three parts: the prompt, the reference information and the output. To simplify this process, we have the following assumptions. (1) the length of prompt and output is the same and fixed for every agent, denoted as $m$. (2) As for the round $k$, we must take the information from the previous rounds into account. So we assume that each agent has additional $|\mathcal{V}^{(k-1)}| \times m$ tokens as its input. (3) Under the same level of difficulty, $|\mathcal{V}^{(i)}| \approx |\mathcal{V}^{(j)}|$ for $\forall i, j \leq k$.

The total cost can be approximately expressed in the following form:

$$\mathcal{C}_{\text{total}} = \sum_i^{|\mathcal{V}^{(k)}|} m + m \times |\mathcal{V}^{(k-1)}| + m \times |Agent_i[\text{ref}]|| + m \times |W_{\text{ref}}(Agent_i)|, \qquad (16)$$

where $W_{\text{ref}}(Agent_i)$ is defined as $\{a \mid Agent_i \in a[\text{ref}]\}$, which contains all agents that have referenced $Agent_i$. This expression can be further simplified to Eq. 17

$$\mathcal{C}_{\text{total}} = m \times (|\mathcal{V}^{(k)}| + |\mathcal{V}^{(k)}| \cdot |\mathcal{V}^{(k-1)}| + \sum_i^{|\mathcal{V}^{(k)}|} (|Agent_i[\text{ref}]| + |W_{\text{ref}}(Agent_i)|)). \qquad (17)$$

Notice that $\sum_i^{|\mathcal{V}^{(k)}|} |Agent_i[\text{ref}]| = \sum_i^{|\mathcal{V}^{(k)}|} |W_{\text{ref}}(Agent_i)| = |E|$, the total cost is given by Eq. 18.

$$\mathcal{C}_{\text{total}} = m \times (|\mathcal{V}^{(k)}| + |\mathcal{V}^{(k)}| \cdot |\mathcal{V}^{(k-1)}| + 2|E|). \qquad (18)$$

With the assumption (3), the average cost for each agent is given by Eq. 19.

$$\bar{\mathcal{C}} = m \times (1 + |V| + 2\frac{|E|}{|V|}). \qquad (19)$$

Notice that topology with linear structure always has lower complexity score. However, the linear structure lacks the ability to call agents in parallel. That means the next agent must wait until current agent finish its task instead of work in the same time. Considering this time cost (also called delay), we take graph depth $d$ into account. When minimizing the average cost, we can ignore the constant part and token length $m$. Then we obtain the expression of topology density before normalization.

$$\mathcal{S} = |V| + 2\frac{|E|}{|V|} + d. \qquad (20)$$

The interaction cost is then analytically linked to three topological features:

- **Number of Agents** $N = |V|$: The total number of agents is a primary driver of base computational and memory overhead. Each agent typically encapsulates a large language model (LLM) or a policy network, thus the cost of inference, state maintenance, and context management scales at least linearly with $N$. This represents the fixed cost of maintaining the system.

- **Edge Density:** The average degree $\bar{e} = \frac{|E|}{|V|}$ correlates with interaction overhead. Higher density implies more pairwise interactions per nodes, increasing synchronization and message-passing costs.

- **Graph Depth** $d$: The number of nodes of the longest path between any two agents defines the worst-case coordination latency. Large depths necessitate multi-hop communications, amplifying delay and potential error propagation.

The number of agents and edge density can be explicitly derived from the definition of the YAML field. However, the depth $d$ needs additional calculations. To cope with this problem, we extract the properties of manager-guided multi-agent interaction and conclude it as the following theorem.

**Theorem 1.** Given DAG $\mathcal{G}^{(k)}$ defined by manager-guided multi-agent interaction, $\mathcal{G}^{(k)}$ is a partite-graph with $b$ parts. Then we have $d^{(k)} = b$, where $d^{(k)}$ is the depth of $\mathcal{G}^{(k)}$.

*Proof.* First, we prove that there exists a path with length $b$, equivalently, **there exists a path that sequentially visits each part** $V_1, V_2, \ldots, V_b$.

By definition, $V_1$ contains only **sources** (no incoming edges from within $\mathcal{G}^{(k)}$), and $V_b$ contains only **sinks** (no outgoing edges within $\mathcal{G}^{(k)}$). Choose any sink $t \in V_b$. Since $t \in V_b$ and edges go from lower to higher parts, $t$ must have a predecessor $p_{b-1} \in V_{b-1}$ (if $b > 1$). Similarly, $p_{b-1}$ must have a predecessor $p_{b-2} \in V_{b-2}$. Repeating this process yields a path backwards from the sink:

$$p_1 \rightarrow p_2 \rightarrow \cdots \rightarrow p_{b-1} \rightarrow t,$$

where $p_i \in V_i$ for $i = 1, 2, \ldots, b-1$. The forward path $P = p_1 \to p_2 \to \cdots \to p_{b-1} \to t$ visits $b$ different parts $(V_1, V_2, \ldots, V_b)$ and contains exactly $b$ vertices.

**Then we prove that $d \leq b$.**

Assume that a path $P = v_1 \to v_2 \to \cdots \to v_m$ exists with $m > b$ vertices. Let $v_i \in V_{a_i}$. Since any edge $v_i \to v_{i+1}$ must satisfy $a_i < a_{i+1}$ (by the Definition 1), the sequence of part indices is strictly increasing:

$$a_1 < a_2 < \cdots < a_m.$$

This sequence has $m$ distinct integers. However, these integers must all lie in the set $\{1, 2, \ldots, b\}$, which contains only $b$ distinct integers. The assumption $m > b$ requires finding more than $b$ distinct integers in a set of size $b$, which is impossible. Therefore, no such path $P$ can exist. Consequently, any path has at most $b$ vertices, and the depth $d \leq b$. □

We must emphasize that in most cases, the agent calling steps satisfy $s = b$, which means $b$ can be directly calculated. However, in rare cases, inter-interactions may not happen between two layers, e.g. $\mathcal{V}_i^{(k)}$ and $\mathcal{V}_j^{(k)}$. In this situation, $\mathcal{V}_i^{(k)} \cup \mathcal{V}_j^{(k)}$ is an independent set, which leads to $b < s$ and additional response time. So we use $s$ as a measurement of the graph depth to recognize the two sequences with the same topology.

Now we have the basic expressions of topology density as Eq. 21

$$\mathcal{S} = |V| + 2\frac{|E|}{|V|} + s. \tag{21}$$

**Topology Density Normalization** With the difficulty level $l$, we have the maximum allowed number of nodes $N_{\max}(l)$. To normalize the density of different difficulties into the same distribution, we scale the formula to (0, 1).

First, we have $\frac{|V|}{N_{\max}(l)} \leq 1$. After limiting the upper bound of $|V|$, we further constrain the limitation of $\frac{|E|}{|V|}$. Notice that the agent communication edges are categorized into three types, *intra-round edges*, *inter-round cross-agent edges* and *inter-round self-edges*. Among them, we have *intra-round edges* $|E_{intra}| \leq \frac{|V|(|V|-1)}{2}$ with the Definition 1 for the intra-round edges. For the inter-round edges, *inter-round self-edges* can be approximately equal to $|V|$ with the assumption (3), and we have *inter-round cross-agent edges* $|E_{cross\_inter}| \leq |V|(|V|-1)$. Then for the edge density,

$$\bar{e} \leq \frac{|E_{intra}|}{|V|} + \frac{|E_{self\_inter}|}{2|V|} + \frac{|E_{cross\_inter}|}{2|V|}, \tag{22}$$

with the simplified form $\bar{e} \leq |V| - 0.5$. Then the normalization form is $\frac{|E|}{|V|(|V|-0.5)}$. When the topology degenerate as linear structure, the depth $d$ is equal to $|V|$ which is the upper bound. So we have $\frac{z}{|V|} \leq 1$.

When complexity gets higher, it requires the final expression of complexity score to decrease. So, we implement a monotonically decreasing activate function in the final expression of the complexity score $\mathcal{S}_{complexity}$ with exponential function $e^{-x}$ in Eq. 7.

### D.3 DETAILED DEFINITIONS OF MULTI-AGENT ROLES

Inspired by the design of MapCoder, our agent pool consists of six distinct agent types, each dedicated to different functions in the code generation process. In each round of code generation, the Managing Agent performs reasoning and selects the necessary agents from this pool. The names and token representations of each agent type are outlined in Figure.2 middle.

#### D.3.1 RETRIEVAL AGENTS

Following Search-R1(Jin et al., 2025), the following retrieval agents employ the E5 model as the unified retriever. E5 serves as the retrieval backbone and is invoked by retrieval agents to identify semantically relevant documents during inference. The retrieval agents can incorporate inputs from other agents as reference context to enhance retrieval accuracy. To enable retrieval of semantically

similar code solutions, we construct an offline retrieval agent. Following VoyageAI, we create a document for each elementary programming problem with a canonical solution (i.e., APPS, HumanEval, and MBPP) by concatenating the description of the natural language problem with its corresponding reference implementation. advanced library usage.

### D.3.2 Planning Agent

The Planning Agent takes as input the original problem along with the outputs of other agents selected by the managed agent in the previous step, and aims to generate a step-by-step coding plan for solving the original problem. In addition, the Planning Agent can iteratively refine its plan based on previous error messages and the last-round plan, aiming to produce a more effective solution strategy.

### D.3.3 Algorithmic Agent

The algorithmic agent takes as input the code problem and the outputs of other agents, and produces a customized sequence of algorithmic solution steps tailored to the given problem.

### D.3.4 Coding Agent

The Coding Agent generates an initial code solution by leveraging the problem description, the step-by-step coding plan produced by the Planning Agent, and reference materials—such as code snippets or tutorials—retrieved by the Retrieval Agent.

### D.3.5 Debugging Agent

Starting from the second round, when the initial code generation encounters issues, the Debugging Agent can iteratively revise the code by leveraging previous error messages and interaction history. Alternatively, it can regenerate code based on the updated coding plan and newly retrieved reference materials. The specific strategy adopted is determined by the Planning decisions made by the Managing Agent.

### D.3.6 Testing Agent

At the end of each iteration, we invoke the Testing Agent to evaluate the correctness of the generated code. It returns a binary pass/fail signal along with graded error diagnostics, which are used both for computing the reward function and as a termination criterion for the iterative process.

## E Supplementary Definitions for RL

### E.1 Definitions of Multi-Turn Trajectories and Returns in RL

We define the multi-turn trajectory as:

$$\tau = \{(o_k,\, z_k,\, r_k)\}_{k=0}^{K-1}, \tag{23}$$

where $o_k$ is the YAML token sequence encoding the interaction topology of turn $k$, $z_k$ denotes the corresponding multi-agent execution outcome produced by the environment, and $r_k$ is the immediate reward assigned based on the execution result. The reward is computed via a function $r_\phi(\cdot)$ that evaluates the current interaction graph and the code validation outcome:

$$r_k = r_\phi\big(\mathcal{G}^{(k)},\, z_k^{\mathrm{code}}\big) \tag{24}$$

where $z_k^{\mathrm{code}}$ is the result of executing input–output test cases in a sandboxed code-validation tool. Different rewards or penalties are assigned depending on whether the code passes the tests or on the specific type of error encountered. In addition, the structural contribution is computed based on whether the topology density of $\mathcal{G}^{(k)}$ stays within a task-specific upper bound determined by the difficulty of the problem. The overall return of a trajectory is defined as the discounted sum of per-turn rewards:

$$R(\tau) = \sum_{k=0}^{K-1} \gamma^k\, r_k, \tag{25}$$

where $\gamma \in [0, 1]$ is a discount factor that modulates the relative importance of earlier versus later rewards. This return serves as the training signal for optimizing the policy.

## E.2 REINFORCEMENT LEARNING OBJECTIVE FOR GENERATING TOPOLOGIES WITH ADAPTIVE COMPLEXITY

The general return $R(\tau)$ serves as the training signal to optimize the topology generation policy, which aims to produce interaction graphs with dynamic structural complexity adapted to the difficulty of the input problem, while maximizing the likelihood of generating code that passes all test cases. Our goal is to maximize expected return on trajectories sampled from the current policy, while regularizing against a reference policy using a token-level Kullback–Leibler (KL) divergence. Notably, the policy $\pi_\theta$ is responsible only for generating the topology token sequences $o_k$; all agent responses, code execution traces (contained in $z_k$) are treated as environment outputs and are excluded from the KL regularization term.

We define the following trajectory-level optimization objective:

$$\max_\theta \ \mathbb{E}_{x \sim \mathcal{D}, \ \{o_k\} \sim \pi_\theta}[R(\tau)] \ - \ \beta \, \mathbb{E}_{\{o_k\} \sim \pi_\theta} \left[ \frac{1}{L(\tau)} \sum_{k=0}^{K-1} \sum_{u=1}^{|o_k|} \log \frac{\pi_\theta(o_{k,u} \mid x, H_k, o_{k,<u})}{\pi_{\text{ref}}(o_{k,u} \mid x, H_k, o_{k,<u})} \right] \quad (26)$$

where $\tau = \{(o_k, z_k, r_k)\}_{k=0}^{K-1}$ is the trajectory induced by the topology sequences $\{o_k\}$ sampled from the policy $\pi_\theta$, with the corresponding interaction graphs, agent outputs, and rewards deterministically generated by the environment. The term $L(\tau) = \sum_{k=0}^{K-1} |o_k|$ denotes the total number of topology tokens in the trajectory, and $\beta$ is a weighting coefficient that balances reward maximization against policy divergence. Here, $x$ is a problem instance drawn from the dataset $\mathcal{D}$, and $o_{k,<u} = (o_{k,1}, \ldots, o_{k,u-1})$ denotes the prefix token sequence generated prior to position $u$ in round $k$.

## E.3 REWARD DESIGN AND SENSITIVITY ANALYSIS

### E.3.1 REWARD DESIGN PRINCIPLES

*Our reward design follows three core objectives: (1) ensuring syntactic validity of the YAML topology, (2) guaranteeing functional correctness of the generated solution, and (3) controlling communication cost by encouraging difficulty-aware sparsity in the agent topology. These objectives are realized through two components: $r_e$ for execution correctness (syntax and solution outcome), and $r_g$ for topology density. The separation enables targeted optimization for both correctness and structural efficiency.*

**YAML Format and Structural Validity.** *Invalid YAML structures receive a strong negative reward, as they cannot support valid multi-agent execution. Other YAML format penalties apply only to the topology structure itself and are independent of roles or tasks. Once the YAML structure is correct, the penalty becomes zero, enabling $r_e$ to focus solely on program execution correctness.*

**Topology Density Reward.** *The density reward $r_g$ consists of three components—the number of nodes, number of edges, and graph depth—as defined in Eq. 7 of the main paper. The weights $(\lambda_1 : \lambda_2 : \lambda_3)$ are not heuristic but derived from the theoretical analysis of information flow in layered DAGs. Following Eq. 20, we adopt $\lambda_1 = 0.5$, $\lambda_2 = 1$, and $\lambda_3 = 0.5$ to preserve the theoretical $1{:}2{:}1$ ratio. This ensures that topology sparsity is accurately captured, while keeping the total reward magnitude comparable to execution-based signals such as code correctness.*

**Difficulty-Aware Density Bounds.** *We additionally set topology density upper bounds of 4, 7, and 10 for tasks of different difficulty levels. These values are obtained through statistical analysis of thousands of SFT-generated samples, examining the distribution of topology densities required for successful solutions.*

## E.4 SENSITIVITY ANALYSIS OF $r_e$ AND $r_g$

*Both reward components are initialized with equal weights $w_1 = w_2 = 1$ for $r_e$ and $r_g$. This balanced configuration reflects their comparable ranges and the need to trade off correctness with communication efficiency. To examine robustness, we perform a sensitivity analysis by sweeping the topology density weight $w_2$ from 0.25 to 2.0, while fixing $w_1 = 1$. Using the APPS dataset, we begin with the post-SFT model and train for 60 steps (50% of total training). For each $w_2$, we record Pass@1 accuracy and relative token cost.*

Table 6: *Sensitivity analysis of the topology weight $w_2$. Results are Pass@1 and relative token cost after 60 RL steps on APPS.*

| $w_2$ (topology weight) | Pass@1 | Rel. Token Cost |
|---|---|---|
| 0.25 | 37.8 | 1.61× |
| 0.50 | 38.0 | 1.46× |
| **1.0 (ours)** | **40.1** | **1.00×** |
| 1.50 | 39.2 | 0.88× |
| 2.00 | 38.9 | 0.76× |

**Observations.** *The results indicate that performance remains stable across a wide range of $w_2$ values. Pass@1 varies by less than 5.2% throughout the sweep (from 38.0 to 40.1), demonstrating that model performance is not sensitive to the precise value of $w_2$. As expected, larger $w_2$ penalizes dense topologies more strongly, resulting in reduced token cost. Importantly, no instability is observed across all settings. Thus, the main conclusions of the paper are robust under reasonable perturbations of reward weights.*

