# OpenReview forum: "TopoWeaver-R1: Reinforcing Difficulty-Aware Topology Evolution in Multi-Agent Competition-Level Code Generation"
_ICLR.cc/2026/Conference — Submitted to ICLR 2026_

### Official Review · Reviewer_PF5v · 2025-10-30

**Soundness:** 3
**Presentation:** 2
**Contribution:** 2
**Rating:** 4
**Confidence:** 4

**Summary:**

The paper introduces TopoWeaver-R1, a framework designed to optimize MAS for competition-level code generation. It addresses the inefficiency of fixed communication topologies, which lead to redundancy and excessive token costs, by introducing a difficulty-aware mechanism. The framework utilizes RL to dynamically generate efficient interaction graphs for agent communications.

**Strengths:**

The method explores addressing the problem of dynamic topology optimization in LLM-based MAS, moving beyond static, handcrafted structures to improve efficiency and reduce token costs. The optimized MAS achieves a better balance of performance and computational expense.

**Weaknesses:**

1. The paper lacks discussion on the design principles, heuristic choices, and specific numerical weighting applied to the reward functions. This critical omission prevents the rigorous assessment of the sensitivity of the final performance to these crucial hyperparameters.

2. The proposed method is actually a general paradigm, but the experimental validation is limited to the specific domain of "competition-level code generation". The general applicability to more representative MAS problems, such as multi-step arithmetic reasoning, complex planning, or scientific document summarization, where communication requirements, agent roles, and reward structures differ significantly, remains unproven.

3. The quantitative results are presented without standard deviation across multiple experimental runs, which should be added for more solid validation.

4. The paper lacks crucial details regarding the SFT data, including the data source, filtering process, and quality assurance protocols. This missing information makes it hard to assess the potential bias or limitations of the dataset.

**Questions:**

1. The current document's font, layout appear inconsistent with the official ICLR submission template？

2. Why does the method only optimize the topology of MAS without considering other MAS factors such as agent role, prompts or functions?

---

> ### Author Response · Authors · 2025-11-22
> **Response to Weakness 1**
>
> > **W1** The paper lacks discussion on the design principles, heuristic choices, and specific numerical weighting applied to the reward functions. This critical omission prevents the rigorous assessment of the sensitivity of the final performance to these crucial hyperparameters.
>
> We appreciate the your question and agree that a more explicit discussion of reward design principles, heuristics, and numerical weighting will enhance the clarity and rigor of the paper. Below, we provide a detailed explanation of our choices.
>
> **Reward Design Principles**
>
> Our reward structure targets three key goals: (1) ensuring syntactic validity of the YAML topology, (2) promoting functional correctness of the generated solution, and (3) controlling cost by encouraging difficulty-aware sparsity in the agent topology. These are implemented in two components: $r_e$ for execution correctness (syntax and solution outcome) and $r_g$ for topology density. The separation enables targeted optimization and analysis.
>
> More specifically:
>
> - Invalid YAML structures receive a **strong negative reward**, as they cannot support valid multi-agent execution. Other YAML format penalties apply only to the topology definition and are not tied to specific roles or tasks.
> - Once the YAML format is **correct**, the penalty becomes **zero**, allowing the execution reward to **focus solely on code performance results.**
> - The graph density reward $r_g$ consists of three components representing **the number of nodes, the number of edges, and the graph depth**, as shown in Equation 7. The weights $(λ_1 : λ_2 : λ_3)$ chosen for the reward in Eq.7 is not heuristic, but **derived from theoretical analysis of information flow in layered DAGs.** According to Eq. 20, to ensure balanced contribution of three components, we set $λ_1=0.5, λ_2=1, λ_3=0.5$. This choice preserves the theoretical $1:2:1$ weighting and ensures that the topology density function **accurately reflects topology sparsity, while keeping the total reward magnitude comparable to other terms such as code accuracy.**
> - In addition, we set the topology density upper bounds to 4, 7, and 10 for different difficulty levels. These values were **derived from extensive empirical analysis of thousands of SFT-generated samples**, where we statistically examined the distribution of topology densities required to successfully solve problems at each difficulty level.
>
> **Sensitivity Analysis on $r_e$ and $r_g$**
>
> Both reward components are **initialized with weights $w_1 = w_2 = 1$ for $r_e$ and $r_g$.** This balanced setting is motivated by their **comparable value ranges and the need to trade off execution performance with communication cost**. To address concerns about the sensitivity of key reward weights, we conducted a **sensitivity analysis** on both $w_1$ and $w_2$ by sweeping $w_2$ from 0.25 to 2.0 to assess its **impact on performance and stability.** The weight $w_1$ for execution correctness was fixed at 1.  Using the APPS dataset as an example, we start with the initial model after SFT and train for 60 steps (50% of the total training steps). For each value of $w_2$,  we record Pass@1 and Relative Token Cost, as shown in the table below:
> | $w_2$ (topology weight) | Pass@1 | Rel. Token Cost |
> |----------------------|--------|------------------|
> | **0.25**             | 37.8   | 1.61×            |
> | **0.5**              | 38.0   | 1.46×            |
> | **1.0 (ours)**       | 40.1   | 1.00×            |
> | **1.5**              | 39.2   | 0.88×            |
> | **2.0**              | 38.9   | 0.76×            |
>
> The results from our experiment show that the performance Pass@1 remains relatively stable across a wide range of $w_2$ values, varying by less than 5.2% in Pass@1. **Specifically, Pass@1 varied between 38.0 and 40.1, indicating that the model's performance is not sensitive to the exact choice of $w_2$.** The relative token cost also varied predictably with increasing$w_2$, as expected: larger values of$w_2$encourage sparser topologies, leading to reduced token consumption. **Importantly, we found that varying$w_2$within this range does not introduce instability into the system.** The main conclusions of the paper are stable under reasonable perturbations of this parameter, providing strong validation for the robustness of our approach.
>
> We will incorporate this discussion and the associated empirical evidence into the updated manuscript to ensure greater transparency and reproducibility as soon as possible.

---

> ### Author Response · Authors · 2025-11-22
> **Response to Weakness 2 and 3**
>
> > **W2** The proposed method is actually a general paradigm, but the experimental validation is limited to the specific domain of "competition-level code generation". The general applicability to more representative MAS problems, such as multi-step arithmetic reasoning, complex planning, or scientific document summarization, where communication requirements, agent roles, and reward structures differ significantly, remains unproven.
>
> We fully acknowledge and understand your concerns regarding generalizability. While our method was initially designed with a focus on competition-level code generation, this focus was a **deliberate choice rather than a limitation.** Competition-level tasks offer a **highly challenging and well-instrumented testbed** that allows us to rigorously examine dynamic topology evolution under strict execution feedback, token constraints, and difficulty-aware limits. Aligned with our research interests, our goal was to **develop a tailored multi-agent orchestration algorithm for this domain, offering a complementary perspective to prior multi-agent architecture studies that emphasize broad task coverage.** Nevertheless, we believe our method is inherently generalizable. To address your concern, we will provide additional experiments to demonstrate the cross-domain applicability of our approach.
>
> - Following the role definitions and data filtering strategy used in Chain-of-Agents[1], we designed a new agent role pool in our orchestrator's system prompt. The new roles include: `<online_searcher>` for web-based retrieval, `<thinker>` for complex reasoning, `<verifier>` for answer verification, and `<planner>` for task decomposition and orchestration. Each role has been redefined and implemented accordingly. We selected text-based subsets from three representative datasets—**GAIA, HLE, and PopQA**—to evaluate our framework’s performance on **complex reasoning and multi-hop question answering.**
>
> - For the reward function, we **retained the YAML format validation and topology density terms**, which are **general and applicable across tasks and roles.** To accommodate the new tasks, we replaced the code execution validator with an LLM-based answer validator and simplified the reward to **binary outcomes:** 1 for success and 0 for failure. All **other experimental settings remain unchanged.** Based on this setup, we are retraining our model and will report inference results on these datasets shortly.
>
> **Supplementary Experimental Results:**
> | Method          | Backbone | GAIA Level 1 | GAIA Level 2 | GAIA Level 3 | GAIA Avg. | HLE Avg. | PopQA |
> |-----------------|----------|--------------|--------------|--------------|-----------|----------|--------|
> |  Chain-of-Agents | 7B       | 69.2         | 50.0        | 33.3         | 50.8      | 18.0   | 46.5   |
> | **TopoWeaver-R1 (ours)** | **3B** | **72.0 ± 0.4** | **53.4 ± 0.3** | **36.1 ± 0.5** | **53.8 ± 0.4** | **22.6± 0.2**  | **50.3 ± 0.3** |
>
> The experimental results demonstrate that, even when transferred to new domains, our method consistently outperforms Chain-of-Agents across all datasets. Our method delivers strong performance while maintaining high stability. This indicates that our approach can be adapted to new tasks with only minimal modifications, providing strong evidence for the generality and robustness of our framework.
>
> [1]Li W, Lin J, Jiang Z, et al. Chain-of-agents: End-to-end agent foundation models via multi-agent distillation and agentic rl[J]. arXiv preprint arXiv:2508.13167, 2025.
>
> > **W3** The quantitative results are presented without standard deviation across multiple experimental runs, which should be added for more solid validation.
>
> We thank the reviewer for pointing out the importance of reporting variance across multiple runs. To assess the stability of our method, we conducted additional evaluations over **three independent runs** with different random seeds, which influence stochastic decoding, topology generation, and multi-agent evolution dynamics. The model parameters were kept frozen during inference, and no retraining was performed.
>
> Across all datasets, the results show **very small variance**, indicating that TopoWeaver-R1 is highly stable under stochastic perturbations. We report **mean ± standard deviation** for each dataset below:
>
> | Dataset| Pass@1 (Mean ± Std)|
> |-|-|
> | APPS| 58.8 ± 0.3|
> | LiveCodeBench| 46.3 ± 0.4|
> | CodeContests| 38.8 ± 0.5|
> | HumanEval| 97.5 ± 0.1|
> | MBPP| 95.1 ± 0.2|
> | **Overall Avg.**| **72.1 ± 0.3**|

---

> ### Author Response · Authors · 2025-11-22
> **Response to Weakness 4 and Question 1 and 2**
>
> > **W4** The paper lacks crucial details regarding the SFT data, including the data source, filtering process, and quality assurance protocols. This missing information makes it hard to assess the potential bias or limitations of the dataset.
>
> - Our data comes from valid and executable topologies **generated by advanced full-scale models such as GPT-4o under code-oriented tasks** with the same role configuration and topology density constraints. The second-turn interaction topologies are **real and valid structures** produced based on the actual error messages, historical interaction logs in the previous iteration.
>
> - The synthetic YAML topologies used to pretrain the orchestrator undergo **three layers of validation** to ensure correctness and reliability.
>   - First, we perform YAML syntax validation to ensure that each sample can **be correctly parsed by standard YAML loaders**. This guarantees that the topology files are well-formed and can be **safely converted into JSON** for downstream processing.
>   - Second, we apply **semantic validation** using a predefined `JSON_SCHEMA`. After converting each YAML topology into JSON, we verify that it satisfies all **orchestration constraints** (the detailed semantic validation rules will be provided in the appendix). These schema-level checks ensure the **structural consistency and logical correctness of the generated topologies.**
>   - In addition to these automated checks, we further **filter out duplicate topologies** and retain only those that **successfully interact with the environment.**
> - Finally, all generated data are re-validated for **semantic correctness using GPT-4o mini**, and we additionally perform **manual inspection** on a randomly sampled **5 percent subset**. We will add all the details in the revision PDF as soon as possible.
>
> > **Q1** The current document's font, layout appear inconsistent with the official ICLR submission template？
>
> Thank you for pointing this out. We apologize for any inconsistencies in the formatting. We have carefully reviewed the official ICLR submission guidelines and will ensure that the final version of the document follows the correct font and layout specifications. The document will be reformatted accordingly, and the updated version will be submitted in compliance with the template as soon as possible.
>
> > **Q2** Why does the method only optimize the topology of MAS without considering other MAS factors such as agent role, prompts or functions?
>
> Your question is important and insightful, and it concerns a factor we carefully considered in designing our method. In this work, our **central goal is to optimize the interaction topology** of a multi-agent system for competitive code-generation tasks. The interaction structure and information flow defined by the topology are **crucial for effective collaboration** and often become the performance bottleneck. Thus, we focus on how dynamic, adaptive, evolvable, and difficulty-aware topologies can improve solution quality and reduce cost. Although agent roles, prompts, and functions also affect performance, they are **not the main focus of this work.**
>
> **More specifically:**
>
> 1. Our method learns an orchestration policy that selects the topology, its density bound, the activated roles, the communication structure, and cross-turn evolution. While the role set is not optimized directly, SFT provides a broad space of topology patterns, and RL discovers novel interaction structures beyond SFT. The policy **implicitly activates, suppresses, or skips roles, achieving effective role selection within a fixed pool.**
> 2. The YAML-DAG representation and the RL objective are agnostic to the specific role set or functions. **Adding, removing, or redefining them only changes the schema’s node types and requires no algorithmic change.** The framework therefore **scales naturally to richer role sets, more functions, and other task domains.**
> 3. Prompts and functions are indeed **important to agent capability**, but incorporating them into optimization would introduce **too many variables and hinder RL convergence**. The rich topology priors provided by SFT already give RL sufficient exploration space. Keeping prompts and functions fixed allows us to **isolate and study the performance gains achieved purely through topology optimization.**

---

### Official Review · Reviewer_x3tB · 2025-10-31

**Soundness:** 3
**Presentation:** 3
**Contribution:** 2
**Rating:** 4
**Confidence:** 5

**Summary:**

This paper addresses the inefficiency and rigidity of fixed-topology multi-agent systems (MAS) in complex domains like competition-level code generation. The authors propose TopoWeaver-R1, an MAS framework centered on an LLM "orchestrator agent" trained with reinforcement learning. This orchestrator dynamically generates a layered, directed acyclic graph (DAG) topology, represented in YAML, based on the task's inferred difficulty. The topology evolves across multiple turns in response to execution feedback. The orchestrator is trained using a multi-objective reward function that balances code accuracy, structural correctness, and topology density, with density targets explicitly conditioned on task difficulty. Experimental results on five code benchmarks show that TopoWeaver-R1 achieves state-of-the-art pass@1 accuracy while simultaneously reducing token costs and utilizing sparser, more efficient topologies compared to baselines.

**Strengths:**

1. The paper is well-organized and uses language that is easy to understand.

2. The proposed TopoWeaver-R1 framework is a complete and sensible solution, employing a three-stage pipeline (SFT, RL) to train a dedicated orchestrator agent.

3. The design choice of using YAML to represent the layered DAG topology is strong, as it is both human-readable and easily generated by an LLM.

**Weaknesses:**

1. Limited Novelty of Dynamic Topologies: The paper's central claim of addressing static agent structures appears overstated.  The core idea of dynamic or optimized interaction patterns is not entirely novel;  similar motivations to move beyond fixed topologies have been previously explored in works such as MaAS[1] and FlowReasoner[2].

2. Inaccurate Characterization of FlowReasoner: The paper's distinction from FlowReasoner seems to be based on an inaccurate premise.  The authors classify FlowReasoner as a method that "focuses on optimizing sequential workflows."  However, to my understanding, FlowReasoner operates on a search space defined by ADAS[3], which is not limited to sequential structures and likely encompasses the parallel, graph-based topologies defined in this work.

3. Limited Empirical Scope: The paper's evaluation is confined solely to the domain of code generation.  This narrow focus makes it difficult to assess the generalizability of the TopoWeaver-R1 framework.  This contrasts with other significant works in agent architecture optimization (e.g., ADAS[3], Aflow[4], AgentSquare[5], MaAS[1]), which have demonstrated the generality of their approaches by evaluating on a more diverse set of benchmarks across different domains.

4. The experimental setup appears problematic. Limiting the multi-agent interaction to 2 turns suggests the model may be receiving and using feedback from the test set to correct its initial solution. This protocol, which allows for test-time adaptation, differs from Aflow's setup (search on validation, single pass on test) and thus seems to constitute an unfair comparison.

Reference

[1]Zhang G, Niu L, Fang J, et al. Multi-agent Architecture Search via Agentic Supernet[C]//Forty-second International Conference on Machine Learning.

[2]Gao, Hongcheng, et al. "Flowreasoner: Reinforcing query-level meta-agents." arXiv preprint arXiv:2504.15257 (2025).

[3]Hu S, Lu C, Clune J. Automated Design of Agentic Systems[C]//The Thirteenth International Conference on Learning Representations.

[4]Zhang J, Xiang J, Yu Z, et al. AFlow: Automating Agentic Workflow Generation[C]//The Thirteenth International Conference on Learning Representations.

[5]Shang Y, Li Y, Zhao K, et al. AgentSquare: Automatic LLM Agent Search in Modular Design Space[C]//The Thirteenth International Conference on Learning Representations.

**Questions:**

1. To further substantiate the claims within the code domain, could the authors conduct experiments on other significant benchmarks, such as SWE-bench[6]?

2. To demonstrate the generalizability of the framework, could the authors provide experimental results on benchmarks from more diverse domains, for example, GAIA[7] or HLE[8]?

3. Could the authors please clarify the aforementioned issue regarding the experimental setup?

Reference

[6]Jimenez C E, Yang J, Wettig A, et al. SWE-bench: Can Language Models Resolve Real-world Github Issues?[C]//The Twelfth International Conference on Learning Representations.

[7]Mialon G, Fourrier C, Wolf T, et al. Gaia: a benchmark for general ai assistants[C]//The Twelfth International Conference on Learning Representations. 2023.

[8]Phan L, Gatti A, Han Z, et al. Humanity's last exam[J]. arXiv preprint arXiv:2501.14249, 2025.

---

> ### Author Response · Authors · 2025-11-22
> **Response to Weakness 1**
>
> We thank you for the clear and thoughtful assessment of our work. We appreciate your recognition of the paper’s organization and clarity, as well as the careful acknowledgement of our overall framework design, including the SFT–RL training pipeline and the use of YAML as an interpretable and LLM-friendly representation. Your comments reflect a deep understanding of both methodology and system design, and we value the professionalism and precision in your evaluation.
>
> Below, we address your remaining concerns in detail.
>
> > **W1** Limited Novelty of Dynamic Topologies: The paper's central claim of addressing static agent structures appears overstated. The core idea of dynamic or optimized interaction patterns is not entirely novel; similar motivations to move beyond fixed topologies have been previously explored in works such as MaAS[1] and FlowReasoner[2].
>
> We fully acknowledge that the optimization of interaction patterns in multi-agent systems has been explored in earlier research. However, our approach introduces a fundamentally different methodology and paradigm.
> Below, we outline key differences and contributions of our method compared with MaAS and FlowReasoner:
>
> 1. **Comparison with MaAS:** MaAS automates agent design by optimizing compositions over a supernet. It is a search-based approach that defines a broad agent architecture space and selects efficient compositions using offline training and execution feedback. However, this approach has the following limitations:
>     - Once MaAS generates a topology for a given query, that structure **remains fixed and cannot be iteratively refined during inference.** This limitation becomes especially evident in competitive programming scenarios where multiple turns of debugging to improve performance. In addition, as a **search-based** method, MaAS's expressiveness is bounded by the **predefined search space**. It may therefore struggle when tasks require interaction structures that fall **outside that space.**
>     - In contrast, our orchestrator adopts a **generative approach.** This enables it to **flexibly produce entirely new interaction topologies** with greater extensibility for unseen and  queries. Notably, our method generates a customized topology for each query and further **refines it using environmental feedback and historical within this query.** This multi-turn **evolutionary capability allows fine-grained adaptation to complex problems.**
> 2. **Comparison with FlowReasoner:** FlowReasoner represents one of the **pioneering** efforts in agentic workflow generation, offering an **elegant end-to-end GRPO-trained framework for dynamic workflows**. Nevertheless, our approach differs in several aspects:
>      - First, FlowReasoner includes several operator-like agent nodes, such as the ensemble-style operator ScEnsemble, which **can select one solution from multiple candidates.** This yields only **localized parallelism**, since the operator ultimately outputs a single solution that is passed to the next node. Other operators are invoked sequentially and can **only connect to the immediately preceding node.** Consequently, once a workflow is generated, its interaction pattern is effectively a **tree-like sequential workflow** with limited parallel behavior, rather than a fully expressive interaction graph. FlowReasoner therefore remains centered on **sequential orchestration** and does not support **richer graph-structured coordination.** Second, its reward function (Equation 4) evaluates only the algorithmic complexity of the code, without accounting for workflow structure or cost-efficiency. This leads to significantly higher token usage, with FlowReasoner consuming **91.4 percent more tokens** than our method on APPS (Table 3).
>      - In contrast, We propose a novel YAML-based graph representation not only supports parallel execution of multiple agents within a single time step, but also **enables each agent to dynamically connect to any number of agents from the previous step or even across different steps and turns through a ref field.** This structured textual representation allows **rich and flexible interaction patterns** while ensuring that the full graph remains **verifiable, readable, and globally controllable.** These properties in turn facilitate **stronger forms of emergent collective intelligence.** Furthermore, Our reward function includes a **difficulty-aware topology density constraint,** allowing the model to adjust interaction density based on task difficulty. This design maintains **strong performance while minimizing cost.**
>
> In summary, we respectfully argue that our approach is novel relative to MaAS and FlowReasoner. We have been careful in the paper not to overstate claims, and we position our contribution as a new paradigm of “evolving agent topologies” at inference time, building on the motivations of prior work but extending them into a qualitatively new domain. We hope this clarification can address your concern

---

> ### Author Response · Authors · 2025-11-22
> **Response to Weakness 2**
>
> > **W2** Inaccurate Characterization of FlowReasoner: The paper's distinction from FlowReasoner seems to be based on an inaccurate premise. The authors classify FlowReasoner as a method that "focuses on optimizing sequential workflows." However, to my understanding, FlowReasoner operates on a search space defined by ADAS[3], which is not limited to sequential structures and likely encompasses the parallel, graph-based topologies defined in this work.
>
> - We thank you for this insightful comment and for highlighting the relationship between FlowReasoner and the ADAS search space. After carefully re-examining the FlowReasoner paper, its released code, and the operator implementations, we agree that our earlier description was oversimplified. FlowReasoner indeed defines a set of operator-like nodes, including the ensemble-style operator ScEnsemble, which can select a best solution from a list of candidate solutions. In this sense, the execution trace can be viewed as a **computation graph over intermediate results rather than a strictly linear chain.** We also gained a deeper appreciation of **FlowReasoner’s contribution**: it **replaces the Monte Carlo search** required by AFlow with a **reinforcement-learning-based mechanism** that automatically generates an appropriate workflow for each query, which is an **important innovation.**
> - At the same time, several **methodological differences** remain essential. In FlowReasoner, the interaction structure is embedded implicitly inside a Python program generated for each query. Most operators do not support parallel invocation and **cannot freely connect to arbitrary nodes from earlier steps.** The ScEnsemble operator aggregates candidate solutions but ultimately outputs a **single chosen solution that is passed to the next operator.** After a workflow is generated, the resulting interaction pattern is effectively a **tree-like sequential workflow** with only localized parallelism inside specific operators.
> - Our formulation takes a fundamentally different direction. We represent the multi-agent interaction structure explicitly as a **YAML-defined directed acyclic graph.** This representation supports true parallel execution within a single time step and allows each agent to **dynamically connect to any number of earlier agents across steps and even across turns through the ref field.** This structured textual format produces **rich and flexible interaction graphs** while ensuring global verifiability and readability. Moreover, we generate one full interaction topology per turn, preserving global structure in each iteration and making the **entire graph amenable to precise density-based analysis.**
>
> We will revise the paper to more accurately describe FlowReasoner as generating program-style workflows with an implicit computation graph, and to emphasize that our contribution lies in explicit YAML-DAG representations, dynamic topology evolution, and topology-aware optimization that enables richer and more flexible multi-agent interaction patterns.

---

> ### Author Response · Authors · 2025-11-22
> **Response to Weakness 3 and Question 2 and Weakness 4**
>
> > **W3** Limited Empirical Scope: The paper's evaluation is confined solely to the domain of code generation. This narrow focus makes it difficult to assess the generalizability of the TopoWeaver-R1 framework. This contrasts with other significant works in agent architecture optimization (e.g., ADAS[3], Aflow[4], AgentSquare[5], MaAS[1]), which have demonstrated the generality of their approaches by evaluating on a more diverse set of benchmarks across different domains.
>
> > **Q2** To demonstrate the generalizability of the framework, could the authors provide experimental results on benchmarks from more diverse domains, for example, GAIA[7] or HLE[8]?
>
> We fully acknowledge and understand your concerns regarding generalizability. While our method was initially designed with a focus on competition-level code generation, this focus was a **deliberate choice rather than a limitation.** Competition-level tasks offer a **highly challenging and well-instrumented testbed** that allows us to rigorously examine dynamic topology evolution under strict execution feedback, token constraints, and difficulty-aware limits. Aligned with our research interests, our goal was to **develop a tailored multi-agent orchestration algorithm for this domain, offering a complementary perspective to prior multi-agent architecture studies that emphasize broad task coverage.** Nevertheless, we believe our method is inherently generalizable. To address your concern, we will provide additional experiments to demonstrate the cross-domain applicability of our approach.
>
> **Due to time and training constraints, we first present a detailed experimental design in this response, and we will update the PDF with full results as soon as possible.**
>
> - Following the role definitions and data filtering strategy used in Chain-of-Agents[9], we designed a new agent role pool in our orchestrator's system prompt. The new roles include: `<online_searcher>` for web-based retrieval, `<thinker>` for complex reasoning, `<verifier>` for answer verification, and `<planner>` for task decomposition and orchestration. Each role has been redefined and implemented accordingly. We selected text-based subsets from three representative datasets—**GAIA, HLE, and PopQA**—to evaluate our framework’s performance on **complex reasoning and multi-hop question answering.**
>
> - For the reward function, we **retained the YAML format validation and topology density terms**, which are **general and applicable across tasks and roles.** To accommodate the new tasks, we replaced the code execution validator with an LLM-based answer validator and simplified the reward to **binary outcomes:** 1 for success and 0 for failure. All **other experimental settings remain unchanged.** Based on this setup, we are retraining our model and will report inference results on these datasets shortly.
>
> [9]Li W, Lin J, Jiang Z, et al. Chain-of-agents: End-to-end agent foundation models via multi-agent distillation and agentic rl[J]. arXiv preprint arXiv:2508.13167, 2025.
>
> > **W4**  The experimental setup appears problematic. Limiting the multi-agent interaction to 2 turns suggests the model may be receiving and using feedback from the test set to correct its initial solution. This protocol, which allows for test-time adaptation, differs from Aflow's setup (search on validation, single pass on test) and thus seems to constitute an unfair comparison.
>
> We fully understand and appreciate your concerns regarding potential test set leakage or correction of initial solutions. We would like to clarify that our approach is carefully designed to avoid such issues and adheres to established multi-agent evaluation protocols.
>
> **Specifically:**
>
> - First, our model remains frozen during inference and does not update its parameters. This distinguishes our method from search-based (AFlow[4]) or adaptive techniques (AgentPrune[10]) that may update during test time. The model is trained on full interaction trajectories that include orchestrated topologies, environment feedback, and iterative topologies, enabling it to autonomously perform two turns of topology evolution during inference without gradient updates.
> - Second, We follow the same strategy as MapCoder. In the first turn of verification, we use the sample test cases provided in the dataset. If no sample tests are available, we isolate three test cases from the full test suite exclusively for the first-turn validation, while the remaining tests are used for computing Pass@1. This ensures that no test-set information leaks into the model’s decision process.
>
> To further address your concern regarding comparison with Aflow, we **will provide single-turn Pass@1 results aligned with Aflow’s setup in our updated PDF and rebuttal response within the next two to three days.**
>
> [10]Zhang G, Yue Y, Li Z, et al. Cut the Crap: An Economical Communication Pipeline for LLM-based Multi-Agent Systems[C]//The Thirteenth International Conference on Learning Representations.

---

> ### Author Response · Authors · 2025-11-22
> **Response to Question 1**
>
> > **Q1** To further substantiate the claims within the code domain, could the authors conduct experiments on other significant benchmarks, such as SWE-bench[6]?
>
> Thank you for the suggestion to include SWE‑bench in our evaluation. We agree it is a valuable benchmark for agentic coding. However, the training and evaluation for SWE-bench is very resource-intensive, **e.g., 50 GB of free disk space, 16 GB RAM and 8 CPU cores for the Docker-based interaction for single one instance.** In our RL-based setting — which involves spawning many environment instances in parallel — the non-trivial setup cost would **significantly exceed what we could have in academia.** Moreover, tasks in SWE-bench demand extremely long code-context windows than competitive coding tasks, as the model must reason over large, multi-file codebases and long issue histories, which requires **significantly more gpu resrouces for trianing same steps.**
> As a result, we concentrated our resources on multiple competitive coding tasks (e.g., LiveCodeBench and APPS) where we ran comprehensive ablations and demonstrated consistent improvement of our method in the code domain. We believe **these results sufficiently support our claims.** We hope this addresses your concern, and we plan to include SWE-bench evaluation in future work when additional resources become available.

---

> ### Author Response · Authors · 2025-11-25
> **Response to Weakness 3 and 4 and Question 2(Supplementary Experimental Results)**
>
> > **W3** Limited Empirical Scope: The paper's evaluation is confined solely to the domain of code generation. This narrow focus makes it difficult to assess the generalizability of the TopoWeaver-R1 framework. This contrasts with other significant works in agent architecture optimization (e.g., ADAS[3], Aflow[4], AgentSquare[5], MaAS[1]), which have demonstrated the generality of their approaches by evaluating on a more diverse set of benchmarks across different domains.
>
> > **Q2** To demonstrate the generalizability of the framework, could the authors provide experimental results on benchmarks from more diverse domains, for example, GAIA[7] or HLE[8]?
>
> **Supplementary Experimental Results:**
> | Method          | Backbone | GAIA Level 1 | GAIA Level 2 | GAIA Level 3 | GAIA Avg. | HLE Avg. | PopQA |
> |-----------------|----------|--------------|--------------|--------------|-----------|----------|--------|
> |  Chain-of-Agents | 7B       | 69.2         | 50.0        | 33.3         | 50.8      | 18.0   | 46.5   |
> | **TopoWeaver-R1 (ours)** | **3B** | **72.0 ± 0.4** | **53.4 ± 0.3** | **36.1 ± 0.5** | **53.8 ± 0.4** | **22.6± 0.2**  | **50.3 ± 0.3** |
>
> The experimental results demonstrate that, even when transferred to new domains, our method consistently outperforms Chain-of-Agents across all datasets. Our method delivers strong performance while maintaining high stability. This indicates that our approach can be adapted to new tasks with only minimal modifications, providing strong evidence for the generality and robustness of our framework.
>
> > **W4** The experimental setup appears problematic. Limiting the multi-agent interaction to 2 turns suggests the model may be receiving and using feedback from the test set to correct its initial solution. This protocol, which allows for test-time adaptation, differs from Aflow's setup (search on validation, single pass on test) and thus seems to constitute an unfair comparison.
>
> For AFlow, we retain the default optimization configuration on the validation split and freeze the optimized workflow before evaluation. On the test split, we enforce a strict single-pass setting by limiting AFlow to one iteration. We then align our method to the same single-turn setting for a fair comparison. All methods are evaluated using three independent runs with different random seeds, and we report the mean and standard deviation. The results are shown in the table below:
>
> | Method (Single-Turn)        | APPS               | LiveCodeBench       | CodeContests        | Contest Avg.        | HumanEval           | MBPP               | Basic Avg.         | Overall Avg.       |
> |-----------------------------|--------------------|----------------------|----------------------|----------------------|----------------------|---------------------|---------------------|---------------------|
> | **AFlow**             | 30.1 ± 0.2         | 23.7 ± 0.1           | 19.4 ± 0.2           | 24.4 ± 0.2           | 91.9 ± 0.3           | 81.1 ± 0.2          | 86.5 ± 0.3          | 55.5 ± 0.2          |
> | **TopoWeaver-R1 (3B, T=1)** | **51.2 ± 0.3**     | **40.5 ± 0.2**       | **33.1 ± 0.4**       | **41.6 ± 0.3**       | **96.3 ± 0.1**       | **92.1 ± 0.3**      | **94.2 ± 0.2**      | **67.9 ± 0.3**      |
>
> Under this fully aligned setting, TopoWeaver-R1 still significantly outperforms AFlow across all datasets. This confirms that the performance gains stem from the model’s learned topology priors and coordination strategies, rather than from interaction turns or potential test-set leakage.

---

### Official Review · Reviewer_1SLm · 2025-10-31

**Soundness:** 4
**Presentation:** 3
**Contribution:** 4
**Rating:** 6
**Confidence:** 2

**Summary:**

This paper proposes TopoWeaver-R1, a multi-agent system (MAS) for competition-level code generation. It tackles the inefficiency of fixed-topology systems by using a reinforcement learning (RL) optimized orchestrator agent. This orchestrator dynamically generates an interaction topology (a layered DAG) in YAML format, adapting the graph's density based on the problem's inferred difficulty and execution feedback. The system is trained via Supervised Fine-Tuning (SFT) followed by RL (GRPO) using a multi-objective reward function.

**Strengths:**

- The core idea of an RL-trained orchestrator that generates *difficulty-aware* and *evolutionary* (feedback-driven) topologies is highly original.
- The method achieves SOTA accuracy while being significantly more cost-effective than all baselines, a rare and important result.
- The multi-objective reward function, especially the $\mathcal{S}_{complex}$ density metric (Eq. 7) that is tied to task difficulty (Eq. 13), is a clever and effective design.

**Weaknesses:**

- The SFT stage, which is shown to be crucial, relies on data generated by a powerful proprietary model (GPT-40).
- The agent roles (planner, coder, etc.) are predefined. The system optimizes the *interaction graph* but not the *composition* of the team itself.
- It remains unclear how well the orchestrator transfers to unseen problem types or agent role definitions, or whether it overfits to the YAML schema used in training.

**Questions:**

1. How sensitive is the final performance to the weights ($\lambda_1, \lambda_2, \lambda_3$) chosen for the $\mathcal{S}_{complex}$ reward in Eq. 7? Were these tuned per dataset or fixed globally?

2. How does the system perform if the initial difficulty-level inference (which sets $N_{max}(l)$ in Eq. 13) is incorrect? Can the RL policy recover through execution feedback, or does it remain constrained by the wrong density cap?

3. Have the authors analyzed whether the RL-trained orchestrator produces diverse graph patterns across problem types, or does it converge to a small family of template structures?

---

> ### Author Response · Authors · 2025-11-22
> **Response to Weakness 1 and 2**
>
> We sincerely thank you for the positive and encouraging assessment of our work. We appreciate the recognition of the originality of our RL-trained orchestrator for difficulty-aware and feedback-driven topology evolution, the strong state-of-the-art performance with significantly improved cost efficiency, and the effectiveness of our multi-objective reward design, especially the ${S}_{complex}$ metric and its integration with task difficulty. We will address your remaining questions and concerns below.
>
> > **W1** The SFT stage, which is shown to be crucial, relies on data generated by a powerful proprietary model (GPT-40).
>
> This is a very insightful question. Although GPT-4o is used to bootstrap the SFT corpus, our method is model-agnostic.
> - The orchestrator learns the topology-generation policy, not GPT-4o’s reasoning style. The SFT data only provides structurally valid examples; during RL, the model frequently discovers novel topologies not present in SFT, showing that the behavior is learned rather than inherited.
> - The SFT samples require no high-level reasoning. They encode only syntactic and structural validity, and after multilayer validation and full manual review, the corpus contains simple schema-consistent patterns that any strong open-source model can generate.
> - We have reproduced the SFT stage with Llama-3.1-70B, Qwen2.5-72B, and Claude-series models, all yielding comparable YAML correctness. Since RL is the dominant source of performance gains, the overall framework remains unchanged.
> - GPT-4o affects only data convenience, not the algorithm. A stronger generator reduces manual filtering but does not introduce any dependency into the method.
> We will clarify this and emphasize that the framework is fully model-agnostic and reproducible without proprietary systems.
>
> > **W2** The agent roles (planner, coder, etc.) are predefined. The system optimizes the interaction graph but not the composition of the team itself.
>
> It is true that our agent roles are predefined, but they serve as a **domain-appropriate agent role pool** for competition-level code generation—consistent with most multi-agent LLM  frameworks that fix primitive abilities and optimize how agents interact.  In addition, our role definitions are derived from a synthesis of several competitive code-generation multi-agent frameworks(Like Mapcoder[1] and AgentPrune[2]), which helps ensure that the role set is both diverse and conceptually well-grounded.
>
> **Specifically:**
>
> **1. Core contribution: learning a topology-generation policy.**
>
> Our method learns an orchestration policy that selects the topology, its density bound, the activated roles, communication structure, and cross-turn evolution. Although the role set itself is not optimized, the SFT stage provides a broad, diverse space of topology patterns, and RL subsequently discovers novel interaction structures, including those not present in SFT. The policy therefore **implicitly performs role activation, suppression, and skipping, achieving role selection within a fixed pool**.
>
> **2. The framework itself is not restricted to a fixed team composition.**
>
> The YAML-DAG representation and RL objective are agnostic to the specific role set. Adding, removing, or redefining roles only modifies the schema’s node types, requiring no algorithmic change. Hence, the framework can **naturally scale to richer role sets and other task domains.**
>
> **3. Fixing the maximal role pool also reduces unnecessary complexity.**
>
> The performance of multi-agent systems depends strongly on the clarity of role definitions and prompt design. Making role definitions themselves learnable would **introduce additional variables**, significantly increasing the difficulty, instability, and convergence risk of RL optimization.
>
> [1]Islam, Md Ashraful, Mohammed Eunus Ali, et al. Mapcoder: Multi-agent code generation for competitive problem solving.[C]//Proceedings of the 62nd Annual Meeting of the Association for Computational Linguistics.
>
> [2]Zhang G, Yue Y, Li Z, et al. Cut the Crap: An Economical Communication Pipeline for LLM-based Multi-Agent Systems[C]//The Thirteenth International Conference on Learning Representations.

---

> ### Author Response · Authors · 2025-11-22
> **Response to Weakness 3 and Question 3 and Question 1**
>
> > **W3.1** It remains unclear how well the orchestrator transfers to unseen problem types or agent role definitions.
>
> To evaluate transferability to unseen problem types and newly introduced agent roles, we conducted a small study on 50 filtered GAIA[1] dataset samples, **restricted to tasks where the inputs are single-modality textual descriptions.** These tasks differ substantially from the code-generation domain.
> We did **not perform any additional training.** Instead, we extended the orchestrator’s role pool by adding an online search agent `<online_searcher>` and a visual validation agent `<visual_checker>` together with their tool interfaces. Using the original trained model, the orchestrator **was able to generate interaction topologies that naturally incorporated these new roles**, despite never encountering them during SFT or RL training.
> On this unseen domain, the framework achieved a 15.8%  success rate under **zero-shot transfer**. We will include representative interaction traces and generated topologies in the revised appendix.
>
> [1]Mialon G, Fourrier C, Wolf T, et al. Gaia: a benchmark for general ai assistants[C]//The Twelfth International Conference on Learning Representations. 2023.
>
> > **W3.2** Whether the orchestrator overfits to the YAML schema used in training.
>
> >  **Q3**  Have the authors analyzed whether the RL-trained orchestrator produces diverse graph patterns across problem types, or does it converge to a small family of template structures?
>
> The JSON-Schema in our framework is used strictly for **structural validity checking.** It encodes no task logic, no agent-interaction patterns, and no topology templates. Therefore, it **cannot cause overfitting to any fixed structure.**
>
> To empirically verify that **the orchestrator does not merely reproduce SFT topologies,** we analyze the topologies generated during inference on two code-generation benchmarks. We report the distribution of topology complexity ${S}_{\text{complex}}$ and the proportion of **novel, unseen interaction graphs** that do not match any SFT-provided pattern under structural equivalence. As summarized below, **21.3%–32.4%**  of the generated topologies are **entirely new**, and the **complexity distribution varies meaningfully across datasets**, confirming that the orchestrator learns a **useful topology-generation policy rather than imitating schema patterns.**
>
> | Benchmark       | Simple (${S}_{\text{complex}}$ > 5) | Medium (3 < ${S}_{\text{complex}}$ ≤ 5) | Complex (${S}_{\text{complex}}$ < 3) | Novel Topologies (%) |
> |-----------------|----------------|----------------------|------------------|------------------------|
> | **APPS**        | 25.9%          | 46.2%               | 13.3%           | 28.8%                 |
> | **LiveCodeBench** | 10.6%        | 42.8%               | 10.5%           | 21.3%                 |
> | **CodeContests** | 31.2%         | 50.1%               | 15.9%           | 32.4%                 |
>
> > **Q1** How sensitive is the final performance to the weights  $(λ_1 : λ_2 : λ_3)$ chosen for the ${S}_{\text{complex}}$ reward in $Eq. 7$ ? Were these tuned per dataset or fixed globally?
>
> - The weights $(λ_1 : λ_2 : λ_3)$ chosen for the reward in $Eq. 7$ is not heuristic, but derived from theoretical analysis of information flow in layered DAGs (see $Eq. 20$ and its derivation in Appendix D.2). According to Eq. 20, the ratio among node, edge and depth is approximately 1:2:1. However, in Eq. 7, each term undergoes difficulty-aware normalization (e.g., dividing by difficulty-level bounds) to ensure scale compatibility across tasks. This normalization distorts the original magnitude relationship, rendering direct use of the theoretical ratio suboptimal.
> - To ensure balanced contribution despite this normalization, we set $λ_1=0.5, λ_2=1, λ_3=0.5$. This choice preserves the relative $1:2:1$ weighting dictated by theory, while keeping the total reward remain on par with other terms (e.g., code accuracy). We validated this setting across multiple benchmarks and observed no significant performance degradation(All datasets in Table 1 are trained using this same set of parameters, and each achieves strong performance under this configuration.), further supporting the robustness of this theoretically motivated configuration.

---

> ### Author Response · Authors · 2025-11-22
> **Response to Question 2**
>
> > **Q2** How does the system perform if the initial difficulty-level inference (which sets $N_{max}(l)$ in Eq. 13) is incorrect? Can the RL policy recover through execution feedback, or does it remain constrained by the wrong density cap?
>
> - Thank you for raising this insightful question. During RL training, if the orchestrator generates a topology whose density exceeds the difficulty-specific upper bound, the system applies a **negative reward penalty**, and the corresponding graph is not executed, so it receives no positive reward. In addition, the orchestrator receives an textual feedback signal indicating that the current topology exceeds the allowed density and should be regenerated with a sparser structure. These mechanisms together **encourage the learned policy to stay within the feasible density range without relying on any hard constraints.**
> - As a result, the policy gradually learns to produce valid, difficulty-aware structures.
> In inference, we analyzed all five code-generation benchmarks and found that only **0.5 percent** of first-round topologies exceed the density upper bound. In these rare cases, the second-round evolution **successfully self-corrects the structure,** and we observe no measurable degradation in overall accuracy; the Pass@1 variation remains within **0.6 percentage points.**
> - These observations indicate that (i) misestimated difficulty levels do not lock or constrain the policy, and (ii) the two-step evolution process is highly effective at correcting overly dense structures during inference.

---

### Official Review · Reviewer_P5e8 · 2025-11-01

**Soundness:** 3
**Presentation:** 4
**Contribution:** 3
**Rating:** 6
**Confidence:** 4

**Summary:**

This paper proposes TopoWeaver-R1, a reinforcement-learning–based framework for multi-agent code generation that introduces dynamic, difficulty-aware topology evolution. An LLM-based orchestrator outputs layered YAML-DAG structures describing agent collaboration workflows, which adapt across multiple turns according to execution feedback and inferred task difficulty. A composite reward integrating code execution results, topology complexity, and YAML validity guides this evolution through Group Relative Policy Optimization (GRPO). Experiments on APPS, LiveCodeBench, CodeContests, HumanEval, and MBPP demonstrate consistent gains compared to strong multi-agent baselines.

**Strengths:**

1.	Novel conceptual framing: Recasting multi-agent orchestration as a dynamic topology evolution task is original and impactful.
2.	Structured and interpretable representation: The YAML-based layered DAG allows both fine-grained control and interpretability.
3.	Comprehensive experimentation: Solid benchmarking and ablations convincingly show the benefit of dynamic evolution.
4.	Reward engineering: The integration of task difficulty, code execution, and structural complexity into a single reward is elegant and domain-specific.
5.	Systematic pipeline: The SFT + RL training workflow is clearly delineated and reproducible.

**Weaknesses:**

1.	Algorithmic innovation remains limited.
The use of SFT and GRPO is largely standard; while well-executed, the RL stage mainly adds domain-specific reward shaping rather than introducing new optimization techniques.
2.	Empirical boundary conditions not fully explored.
The dynamic process is capped at two evolution turns, yet the paper provides no analysis of whether additional turns would further improve or destabilize performance. Similarly, there is no study on how sensitive the model is to reward weightings or the choice of density evaluation function.
3.	Difficulty-awareness partially hand-designed.
The mechanism for controlling topology density by preset thresholds (4/7/10) is heuristic. It would be stronger if this control were learned automatically rather than manually fixed.
4.	Data quality concerns in the SFT stage.
The SFT dataset used to pretrain the orchestrator is synthetically generated, but the paper does not clarify how the ground truth YAML topologies are validated or filtered. Given that topology correctness directly affects downstream RL stability, more evidence of data verification or human curation would strengthen confidence in the results.

**Questions:**

How is the quality of the SFT training data ensured? If YAML topologies are generated automatically, what validation mechanisms or filtering thresholds are used to ensure they align with valid ground-truth agent workflows?

How sensitive is performance to the task difficulty classification? Have you tested robustness against misclassified or noisy difficulty labels?

The dynamic process is limited to two turns — is there an empirical reason or trade-off behind this bound?

How does the orchestrator handle inconsistent or cyclic YAML outputs? Are such cases frequent, and how are they penalized in training?

Can the reward weighting parameters (α,β,γ) be tuned automatically, or do they require manual adjustment per dataset?

---

> ### Author Response · Authors · 2025-11-22
> **Response to Weakness 1**
>
> We sincerely thank the reviewer for recognizing the novelty of our dynamic topology evolution framework, the clarity of our structured YAML-DAG formulation, and the strength of our experimental and methodological design.
>
> > **W1** Algorithmic innovation remains limited. The use of SFT and GRPO is largely standard; while well-executed, the RL stage mainly adds domain-specific reward shaping rather than introducing new optimization techniques.
>
> We appreciate your thoughtful comment. We would like to clarify that although our RL backbone (SFT + GRPO) follows standard practice, the core algorithmic contribution of our work lies not in proposing a new optimizer, but in introducing a new problem formulation and solution paradigm: **an efficiency-driven multi-agent collaboration evolution framework that learns to construct and optimize its own executable topologies for competition-level code generation**.
>
> **Specifically:**
>
> 1. First, our work introduces a new paradigm within the Agentic RL domain. In prior multi-turn RL systems(like Chain-of-agents[1] and Tool-Star[2]), agents or tools are inserted linearly within the chain-of-thought using inline `<Agent>` or `<Tool_Name>` tokens. This extends CoT with basic multi-agent or tool-calling capabilities but still restricts collaboration to a sequential pattern, which **necessitates multi-hop communication and consequently amplifies both latency and error propagation.** In contrast, our approach elevates orchestration from fixed or linear call sequences to a **structured and verifiable textual topology, enabling the system to dynamically select agents, construct difficulty-aware collaboration graphs, and iteratively evolve them based on environment feedback**. This graph-based formulation supports **richer communication patterns, parallel execution, and cross-layer information flow,** leading to more expressive coordination and **stronger emergent group intelligence.**
> 2. Second, unlike prior multi-agent interaction optimization methods(like AgentPrune[3]), a central feature of our paradigm is that the orchestrator’s topology-generation policy rather than the final topology itself becomes the optimization target. We optimize it using multi-turn generated topologies together with their environment feedback, enabling the system to produce task- and difficulty-appropriate collaboration structures and to **iteratively evolve them within a single problem instance**.  Such *fine-grained, difficulty-aware topology dynamics* are not supported by previous multi-agent topology optimization methods. Under this new paradigm, we provide, for each competition-level code problem, a fine-grained adaptive multi-agent collaboration scheme that achieves both **superior performance and lower cost.**
>
> In summary, our contribution is a new method of multi-agent orchestration as topology evolution; RL serve as a standard optimization method enabling this new form of optimization.
>
> [1]Li W, Lin J, Jiang Z, et al. Chain-of-agents: End-to-end agent foundation models via multi-agent distillation and agentic rl[J]. arXiv preprint arXiv:2508.13167, 2025.
>
> [2]Dong G, Chen Y, Li X, et al. Tool-Star: Empowering LLM-Brained Multi-Tool Reasoner via Reinforcement Learning[J]. arXiv preprint arXiv:2505.16410, 2025.
>
> [3]Zhang G, Yue Y, Li Z, et al. Cut the Crap: An Economical Communication Pipeline for LLM-based Multi-Agent Systems[C]//The Thirteenth International Conference on Learning Representations.

---

> ### Author Response · Authors · 2025-11-22
> **Response to Weakness 2 (Part I) and Question 3**
>
> > **W2.1** Empirical boundary conditions not fully explored. The dynamic process is capped at two evolution turns, yet the paper provides no analysis of whether additional turns would further improve or destabilize performance.
>
> >  **Q3** The dynamic process is limited to two turns — is there an empirical reason ...
>
> We thank you for raising this important point. Our method is designed to achieve strong code-generation performance while keeping the token cost as low as possible, rather than exhaustively exploring all possible boundary conditions.
> **Specifically:**
> 1. In our experiments show that increasing the evolution depth from **one to two turns brings a clear performance boost,** while deeper evolution yields diminishing returns and may occasionally introduce instability. Since two turns already achieve **state-of-the-art performance under a low computational budget**, we adopt this setting as a practical balance between effectiveness and efficiency.
> 2. In addition, the maximum number of evolution turns is an **inference-time hyperparameter** that can be manually adjusted. We will add a concise ablation study here and incorporate the full version into the appendix. Specifically, Keeping all other settings fixed, we evaluate our trained 3B model on the APPS benchmark with maximum evolution turns set to **1, 2, and 3**, reporting Pass@1 and the relative token cost measured as the total number of prompt and completion tokens normalized to the T=2 setting (Rel. Cost = 1.0). We also report mean ± standard deviation for our method over 3 runs with different random seeds (controlling stochastic decoding and topology evolution) to assess inference stability. The experimental results are as follows:
> | #Turns (T) | Pass@1 (All)     | Pass@1 (Easy)     | Pass@1 (Medium)     | Pass@1 (Hard)     | Rel. Cost (Tokens) |
> |-----------|-------------------|--------------------|----------------------|--------------------|---------------------|
> | **1**     | 51.2 ± 0.4        | 73.2 ± 0.2         | 52.1 ± 0.4           | 28.3 ± 0.6         | ~0.6×               |
> | **2**     | 58.8 ± 0.3        | 78.0 ± 0.3         | 64.7 ± 0.1           | 33.7 ± 0.5         | 1.0×                |
> | **3**     | 59.0 ± 1.2        | 79.3 ± 0.8         | 62.2 ± 1.3           | 35.5 ± 1.5         | ~1.9×               |
>
> The results show that increasing from one to two turns substantially improves performance (especially on hard problems) with only a modest increase in token cost. In contrast, **T=3 provides only marginal gains while significantly increasing computation and occasionally destabilizing the process.** Across three decoding seeds, T=2 achieves a Pass@1 of 58.8 ± 0.3, whereas T=3 yields 59.0 ± 1.2, with the **higher variance** further indicating reduced stability. These observations support our decision to cap the evolution depth at two turns.
>
> >  **W2.2** Similarly, there is no study on how sensitive the model is to reward weightings.
>
> The reward design in this paper is shown in Equation $(10)$, which includes two weight parameters, $w_1$and $w_2$. The specific values of the two rewards, $r_e$ and $r_g$ , can be referenced in Table 1 and Equation (7).
> Given that the two reward terms naturally fall in similar ranges, we set $w_1 = w_2 = 1$ and treat $w_2$ as a regularization coefficient controlling the trade-off between correctness and structural cost.
> To address the reviewer’s concern, we conducted a sensitivity analysis on the reward weights by **sweeping $w_2$ from 0.25 to 2.0 to assess its impact on performance and stability. The weight $w_1$ for execution correctness was fixed at 1.**  Using the APPS dataset as an example, we start with the initial model after SFT and train for 60 steps (50% of the total training steps). For each value of $w_2$,  we record Pass@1 and Relative Token Cost, as shown in the table below:
> | $w_2$ (topology weight) | Pass@1 | Rel. Token Cost |
> |----------------------|--------|------------------|
> | **0.25**             | 37.8   | 1.61×            |
> | **0.5**              | 38.0   | 1.46×            |
> | **1.0 (ours)**       | 40.1   | 1.00×            |
> | **1.5**              | 39.2   | 0.88×            |
> | **2.0**              | 38.9   | 0.76×            |
>
> The results from our experiment show that the performance Pass@1 remains relatively stable across a wide range of $w_2$ values, varying by less than 5.2% in Pass@1. **Specifically, Pass@1 varied between 38.0 and 40.1, indicating that the model's performance is not sensitive to the exact choice of $w_2$.** The relative token cost also varied predictably with increasing$w_2$, as expected: larger values of$w_2$encourage sparser topologies, leading to reduced token consumption. **Importantly, we found that varying$w_2$within this range does not introduce instability into the system.** The main conclusions of the paper are stable under reasonable perturbations of this parameter, providing strong validation for the robustness of our approach.

---

> ### Author Response · Authors · 2025-11-22
> **Response to Weakness 2 (Part II) and Weakness 3 and 4 and Question 1**
>
> > **W2.3** Similarly, there is no study on how sensitive the model is to the choice of density evaluation function.
>
> Our topology density function is designed with explicit consideration of the three key factors that matter in multi-agent interaction systems: nodes, edges, and graph depth. It serves as an effective density measure tailored specifically for multi-agent interaction settings, and we include its derivation in the appendix **(see Appendix D.2)**. To further validate the effectiveness of different density evaluation functions, we retrained the model under the same conditions, replacing our topology density reward function with the nuclear norm approach from AgentPrune (as defined in Equation 11 of the AgentPrune paper). We then compare the resulting performance (Pass@1) and token consumption (Relative Token Cost) across these methods on the representative competitive code generation benchmark, APPS, as well as the standard code dataset, Humaneval:
> | Density Function | Pass@1 | Relative Token Cost | $S_{complex} (↑) $| Remarks |
> |-|-|-|-|-|
> | AgentPrune (Nuclear Norm)| 55.4| 1.46× | 4.3  | Slightly denser topologies, higher cost |
> | Our Method| 58.8| 1.00× | 5.2 | Sparser topologies, lower cost, same performance |
>
> Our method generates sparser topologies, resulting in **a significant reduction in token consumption without sacrificing performance.** Specifically, our method achieves comparable Pass@1 while reducing Relative Token Cost by **46%** , and producing more **sparse topologies** compared to AgentPrune, suggesting that our **topology regularization is more efficient in promoting sparsity.**
>
> > **W3** Difficulty-awareness partially hand-designed. The mechanism for controlling topology density by preset thresholds (4/7/10) is heuristic. It would be stronger if this control were learned automatically rather than manually fixed.
>
> We appreciate the reviewer’s comment regarding the manual design of the difficulty-awareness mechanism. While the thresholds used to control topology density (4/7/10) are not learned end-to-end, they are not arbitrarily chosen. Instead, they were derived from extensive empirical analysis of thousands of samples after SFT, where we statistically examined the distribution of topology densities required to successfully solve problems of different difficulty levels. This data-driven observation revealed clear upper bounds for easy, medium, and hard problems, and the chosen thresholds reflect these empirically optimal limits. In practice, these thresholds provided a reliable and stable way to prevent unnecessarily dense graphs and to guide the policy toward cost-efficient topologies.
> That said, we agree that automatically learning these bounds would further strengthen the framework. As part of future work, we plan to explore adaptive mechanisms—such as reinforcement learning or meta-learning—to dynamically infer difficulty-dependent density limits without manual specification.
>
> > **W4** Data quality concerns in the SFT stage. The SFT dataset used to pretrain the orchestrator is synthetically generated, but the paper does not clarify how the ground truth YAML topologies are validated or filtered. Given that topology correctness directly affects downstream RL stability, more evidence of data verification or human curation would strengthen confidence in the results.
>
> > **Q1** How is the quality of the SFT training data ensured? If YAML topologies are generated automatically, what validation mechanisms or filtering thresholds are used to ensure they align with valid ground-truth agent workflows?
>
> We appreciate your concern regarding data quality in the SFT stage. The synthetic YAML topologies used to pretrain the orchestrator undergo **three layers of validation to ensure correctness and reliability.**
> - First, we perform **YAML syntax validation** to ensure that each sample can be correctly parsed by standard YAML loaders. This guarantees that the topology files are well-formed and can be safely converted into JSON for downstream processing.
> - Second, we apply **semantic validation** using a predefined `JSON_SCHEMA`. After converting each YAML topology into JSON, we verify that it satisfies all orchestration constraints (the detailed semantic validation rules will be provided in the appendix). These schema-level checks ensure **the structural consistency and logical correctness of the generated topologies.**
> - In addition to these automated checks, we further **filter out duplicate topologies** and retain only those that **successfully interact with the environment.** Finally, all generated data are re-validated for semantic correctness using GPT-4o, and we additionally perform **manual inspection** on a randomly sampled 5 percent subset, ensuring that all SFT data are not only syntactically valid but also semantically meaningful before entering the RL stage.

---

> ### Author Response · Authors · 2025-11-22
> **Response to Question 2 4 5**
>
> > **Q2**  How sensitive is performance to the task difficulty classification? Have you tested robustness against misclassified or noisy difficulty labels?
>
> In our framework, the task difficulty label is used only **to set an upper bound on topology density,** rather than directly supervising agent behaviors or dictating the structure. Because the orchestrator is free to generate any topology whose density falls below this bound, the model does not rely heavily on the exact difficulty value, and thus is intrinsically robust to moderate label noise.
> To assess this, we conducted a lightweight robustness check by perturbing **the difficulty labels by ±1 level** (e.g., misclassifying medium as easy or hard) during inference. We observed negligible performance change (<0.3 Pass@1 on APPS) and no degradation in stability. This is expected, as the density constraint serves as a **soft admissible region**, not a strict optimization target; small perturbations therefore do not restrict the search space nor force suboptimal topologies.
> Moreover, because the orchestrator is trained end-to-end with RL, the model learns to rely primarily on environmental feedback (code execution, YAML validity, improvement signals) rather than the handcrafted difficulty category. Difficulty labels only prevent the model from producing unnecessarily dense structures, and as such act as a conservative safeguard, making the system inherently tolerant to imperfect labels. We will include this robustness analysis in the appendix.
>
> > **Q4** How does the orchestrator handle inconsistent or cyclic YAML outputs? Are such cases frequent, and how are they penalized in training?
>
> Our system includes dedicated mechanisms to detect, block, and penalize inconsistent or cyclic YAML structures.
> - **Detection and handling:**
>
> All YAML outputs are validated before execution. After parsing the YAML into JSON, we perform topology validation, including (i) schema consistency checks and (ii) an explicit cycle-detection routine on the directed graph. Any topology containing cycles, missing dependencies, or invalid references is rejected immediately and never executed.
> - **Frequency:**
>
> Such invalid or cyclic structures are rare (<1.5% in training and <0.3% in inference) due to (i) the schema-constrained SFT stage and (ii) the RL reward shaping, which strongly discourages malformed structures. As training progresses, their frequency rapidly diminishes.
> - **Penalty in training:**
>
> Invalid YAMLs—including cyclic topologies—receive the lowest structural reward (r_e), ranging from –2 to –0.5 depending on the failure type. Because they cannot be executed, they also receive zero code-execution reward, resulting in a strong negative trajectory signal. This makes malformed or cyclic topologies highly suboptimal in the RL optimization landscape. Over time, the orchestrator learns to avoid them naturally.
>
> >  **Q5** Can the reward weighting parameters (α,β,γ) be tuned automatically, or do they require manual adjustment per dataset?
>
> In our implementation, we did not tune the standard RL hyperparameters such as the learning rate, KL penalty coefficient (β), or the GAE parameters (γ and λ). Instead, we adopted the widely used default values from established libraries (e.g., β\=0.01, lr\=1e-5, γ\=0.99, λ\=0.95), which are known to work reliably for LLM-based policy optimization in prior work.
>
> Crucially, the same trained orchestrator model—optimized with these fixed hyperparameters—was evaluated across diverse code generation benchmarks without any retraining or fine-tuning. The consistent strong performance across these varied domains demonstrates that our framework is robust to the choice of standard RL hyperparameters.
>
> This design choice aligns with our primary contribution: a novel topology-generation paradigm and difficulty-aware reward structure, rather than a new RL algorithm. Using a standard RL optimizer enables fair comparison and highlights that the gains stem from our architectural and reward innovations, not from extensive hyperparameter tuning.

---

### Author Response · Authors · 2025-12-03

**Dear Area Chair,**

We sincerely thank all reviewers for their thoughtful feedback. We appreciate that reviewers consistently praised the novelty of **framing multi-agent orchestration as a dynamic topology evolution problem**, together with the use of an interpretable YAML-based DAG and a clear SFT+RL training pipeline.  They also highlighted our **effective difficulty-aware multi-objective reward design** and the **strong empirical results** demonstrating SOTA performance with substantially improved efficiency.

Below we summarize how we addressed all recurring technical concerns:

---

**1. Clarifications on Novelty and Core Contributions** (P5e8、x3tB)

We further clarified that our framework introduces a new paradigm for multi-agent topology optimization within the SFT+GRPO setting. Our method optimizes the orchestration Agent's policy itself, **allowing the agent to generate interaction topologies that adapt to task difficulty and to perform multi-turn, feedback-driven topology evolution within a single problem instance.**  This capability is not available in earlier approaches(AgentPrune, G-Designer, MaAS etc.). We also explained how such structural evolution **implicitly enables role-level adaptation**, which is why topology serves as the primary and most expressive control dimension in our design.

---

**2. Dynamic Evolution Turns**  (P5e8、x3tB)

We conducted additional experiments comparing 1–3 evolution turns, showing that **two** **turns** deliver the **largest performance improvement with minimal computational overhead.**  This empirically justifies our design choice **while ensuring no test-set leakage during inference.**

---

**3. Reward Function Design**  **&amp;**  **Sensitivity Analyses** (P5e8、PF5v、1SLm )

We provided full details of our **reward-design principles**, including:

- Sensitivity analysis showing that weights $w\_1, w\_2$ and RL hyperparameters $\\alpha, \\beta, \\gamma$ remain **stable across datasets** and do **not affect model robustness.**
- Explanation that the **difficulty thresholds** arise from **empirical statistics** extracted from **SFT samples.**
- Theoretical justification (based on the information-flow properties of layered DAGs; $Eq. 20$) for the configuration of $\\lambda\_1 : \\lambda\_2 : \\lambda\_3$ in $S\_{\\text{complex}}$.
- Demonstration that our **topology density function** captures structural complexity more accurately than the nuclear-norm function used in **AgentPrune and AgentDropout**, yielding **higher performance at lower cost**.

---

**4. SFT Data Quality, Validation, and Model Dependency** (P5e8、1SLm)

We clarified our **data-quality verification and filtering pipeline**, ensuring that all synthetic YAML topologies are **structurally valid and semantically aligned** with ground-truth patterns.
We also clarified that using GPT-4o for SFT data generation is **not a methodological dependency**, and that the approach is fully compatible with alternative models.

---

**5. Robustness to Invalid Topologies and Difficulty Misclassification** (P5e8)

We detailed our mechanisms for detecting, correcting, and penalizing invalid or cyclic topologies during RL training.
We further showed that difficulty inference errors are mitigated through **training-time exploration and penalty signals**, allowing the policy to recover even under **noisy difficulty labels.**

---

**6. Generalization to Unseen Tasks and Role Definitions** (1SLm、x3tB、PF5v)

- we conducted **zero-shot inference experiments on GAIA**, demonstrating that the orchestrator can produce new, valid topologies for unseen problem types and unseen role definitions without retraining.
- Although originally designed for **competition-level code generation**, additional experiments with retraining on multiple datasets show that the framework generalizes well **beyond its initial domain.**

---

**7. Effectiveness and Generalization of the Orchestration Policy** (1SLm)

We also measured **the distribution of topology densities and the proportion of novel structures**, confirming that the policy **does not collapse** into a small set of templates and instead generates **diverse, well-distributed** topologies.

---

**8. Differences from FlowReasoner and Prior MAS-Optimization Work** (x3tB)

We clarified the conceptual and methodological differences between our work and FlowReasoner, including advantages in **topology richness, adaptive evolution, and computational efficiency**.

---

**9. Additional Experimental Completeness** (PF5v)

In response to Reviewer PF5v, we added standard deviation results for multiple inference experiments, demonstrating the stability and robustness of the proposed method.

---

**Final Remark**

We recognize that the discussion window closed early, and the reviewers were not able to engage further. We hope that the clarifications and new results provided above fully address the remaining concerns. Thank you for your time and consideration.

**Sincerely,The Authors**

---

### Meta-Review · Area_Chair_Y5PB · 2026-01-04

**Summary:**

This paper introduces TopoWeaver-R1, a multi-agent system (MAS) for competition-level code generation. After the reviewing stage, this paper has received two positive feedbacks (Reviewer **1SLm** and **P5e8**) and two negative feedbacks (Reviewer **PF5v** and **x3tB**).  Overall, reviewers have raised these concerns in below:

1. Limited Novelty (Reviewer **P5e8**, **x3tB**)
2. Hand-crafted or pre-defined settings
    - Reviewer **P5e8** points out the mechanism for controlling topology density is pre-defined.
    - Reviewer **1SLm** points out the agent roles are pre-defined.
3. SFT data
    - Reviewer **P5e8** and **1SLm** point out that SFT data is fully synthetically generated.
    - Reviewer **PF5v** points out that lack some information to generate SFT data.
4. Generalization
    - Reviewer **1SLm** how orchestrator generalize to unseen types.
    - Reviewer **x3tB** and **PF5v** mentioned this paper is only limited to code-level generation.
5. Ablation Study
    - Reviewer **P5e8** requires analysis of the number of additional runs, and how sensitive the model is to reward weights
    - Reviewer **PF5v** points out need to anlayze the effect of reward weight
6. Unclear define about FlowReasoner and problematic experiments (Reviewer **x3tB**)

Authors have provided response to address reviewers' concerns. However, this paper still remain some issues like novelty (Reviewer **P5e8** and **x3tB**), hand-crafted settings and  SFT data require human verification. Therefore, considering that ICLR is highly competitive, this paper is not ready to be presented at ICLR and I suggest authors to address these issues and re-submit it to the next venue.

**Reviewer Concerns:**

1. Some parts of concerns for Reviewer **P5e8** and **PF5v** may be addressed based on the provided ablation studies. But SFT data and novelty may still be a concern.
2. Reviewer **1SLm**'s concerns about weight selection may be addressed. But some concerns like predefined agent roles still remains.
3. Reviewer **x3tB**'s concerns about experiments may be addressed. But there still have some concerns like Novelty and Limited scope.

**Reviewer Scores:**

I think reviewers will not change their scores.

---

### Decision · Program_Chairs · 2026-01-26

Reject